# Cognitive computational model reveals repetition bias in a sequential decision-making task
Eric Legler [1] ✉, Darío Cuevas Rivera [1,2], Sarah Schwöbel[1], Ben J. Wagner [1] & Stefan Kiebel [1,2]

Humans tend to repeat action sequences that have led to reward. Recent computational models, based on a long-standing psychological theory, suggest that action selection can also be biased by how often an action or sequence of actions was repeated before, independent of rewards. However, empirical support for such a repetition bias effect in value-based decision-making remains limited. In this study, we provide evidence of a repetition bias for action sequences using a sequential decision-making task ($N = 70$). Through computational modeling of choices, we demonstrate both the learning and influence of a repetition bias on human value-based decisions. Using model comparison, we find that decisions are best explained by the combined influence of goal-directed reward seeking and a tendency to repeat action sequences. Additionally, we observe significant individual differences in the strength of this repetition bias. These findings lay the groundwork for further research on the interaction between goal-directed reward seeking and the repetition of action sequences in human decision making.

More than a century ago, Thorndike[1] proposed the law of effect, which states that actions that lead to rewarding outcomes are more likely to be repeated. The law of effect gained widespread recognition and is considered an important foundation for the development of early operant conditioning[2] and modern-day reinforcement learning (RL)[3]. What is less known is that Thorndike additionally stated the law of exercise, also known as the law of use, stating that humans tend to repeat previous actions regardless of reward[1].

Consider, for instance, the morning routine that many of us follow, e.g., we start with a cup of tea or coffee, take a shower, have breakfast, brush our teeth, and get ready for work. Although such action sequences may be learned only by goal-directed reward seeking (law of effect), such learning might also be based on repeating actions (law of exercise). Indirect empirical evidence for the law of exercise, i.e., a measurable repetition bias, stems from questionnaire studies on everyday behavior[4–9], showing that participants reliably repeat behavior in a context-dependent manner, for example a specific morning routine or the mode of transportation to work.

Experimental evidence, across disciplines, shows that repetition affects human decision making and learning. It improves language learning[10], likely through increased word-familiarity[11] and better learning of multiword expressions[12]. Repetition also modulates learning of motor and cognitive skills[13–17] and affects memory retrieval, judgment[18,19], and working memory processes[20], demonstrating its broad influence on cognitive functions.

Effects of repetition have also been studied under specific experimental conditions, suggesting their independence from direct reward. Here, repetition significantly affects perceptual decision making by accelerating response times for ambiguous stimuli[21]. Similarly, repetition can alter preferences in value-based decision-making processes[22,23], suggesting that the influence of repetition extends beyond the direct anticipation or receipt of reward, challenging standard views on the effect of reward on decision making and learning. Most importantly, repetitions are seen as a key element of habit formation[24–27].

Over the last decade, the study of habitual vs. goal-directed responses have been enriched through a broad range of studies using devaluation tasks, extinction tests or more complex cognitive tasks, like the Wisconsin card-sorting task[28] or the two-step task[29]. These studies helped broaden our understanding of whether an action is outcome-oriented via a probabilistic map of the environment or rather driven by having obtained past reward in the same situation, as for example might be the reason for an insensitivity to devaluation. For instance, for the two-step task, behavior is described by using a mixture of model-based (MB) and model-free (MF) RL. Here, the MB controller learns a probabilistic action-outcome mapping, i.e., a more sophisticated goal-directed higher order cognitive process, while the MF controller is governed by simpler stimulus-response associations[29,30]. This approach provided many insights on how humans learn and perform tasks[29,31–35], and also highlighted how impairments in MB planning can be

[1]Chair of Cognitive Computational Neuroscience, Faculty of Psychology, TUD Dresden University of Technology, Dresden, Germany. [2]Centre for Tactile Internet with Human-in-the-Loop (CeTI), TUD Dresden University of Technology, Dresden, Germany. ✉e-mail: eric.legler@tu-dresden.de

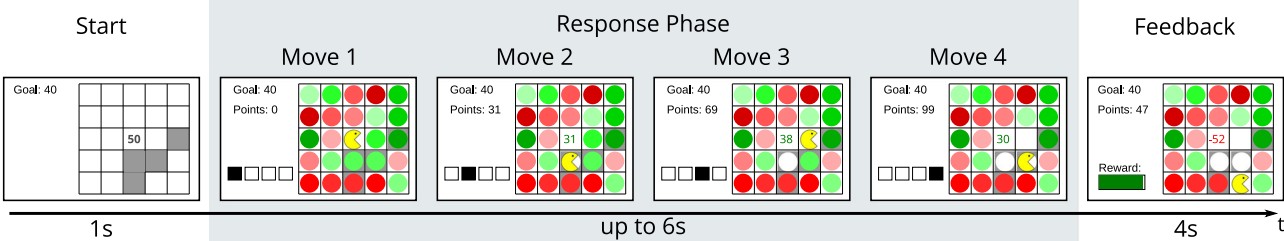

**Fig. 1 | Illustration of a single task trial.** At trial start, the goal, the fields of the default action sequence (DAS) and the mean points of the DAS were shown on the screen for one second. During the subsequent response phase of up to six seconds, four moves had to be performed. During the feedback phase of 4 seconds at the end of each trial, the reward was communicated.

linked to psychiatric disease[36–40]. However, it is still debated whether the reward-driven nature of MF RL aligns with the concept of habits, which is not related to immediate reward, but to mere repetition of actions[26,27]. Two recent studies[41,42] proposed a different repetition-based mechanism. In these studies, based on simulations, behavior was explained by the interaction of two components. First, as usual, goal-directed behavior was explained by a MB planner. Second, the proposal was to model the effect of a repetition bias, following Thorndike's law of exercise, based on past choice counts alone, without regard to outcome or reward. This perspective is also related to minimizing the complexity of an action policy over time[43].

Here, we followed this theoretical lead and assessed empirically, in human participants ($n = 70$), the effect of such a repetition bias on behavior. Our goal was to investigate the mutual influence of rewards and repetition on task behavior. Note that our goal here was not to test hypotheses about habit formation. We used a computational model to disambiguate the effects due to repetition bias from effects due to goal-directed behavior driven by reward maximization. To capture both types of effects, we developed the Y-navigation task (Y-NAT) in which participants perform sequences of movements in a 5-by-5 grid-world to collect a trial-specific number of points. The task was designed to fulfill the following three main objectives: First, trial-specific points establish a clear goal that will prompt goal-directed behavior in participants. Second, the combination of a relatively tight deadline and the requirement to plan four moves ahead (see "Methods") challenges participants in their capacity to act in a purely goal-directed fashion. Third, participants were informed about a so-called default action sequence (DAS), providing them with a less complex go-to strategy, which (1) induces repetition of the same sequence over trials and (2) implicitly signals to highly motivated participants that the task does not require them to always search for the best action sequence but selecting a suboptimal action sequence is considered acceptable task behavior. The Y-NAT therefore enabled us to test (1) whether a repetition bias develops over time, not only for the DAS but for any other action sequence, (2) what its effects are on task behavior, and (3) what the link is between individual differences in repetition bias and overall task performance. Data were analyzed using both standard behavioral analyses and Bayesian model-based analyses. We used Bayesian model comparison to test several alternative models, with or without repetition bias.

## Methods
### Participants
Participants were recruited using the recruitment system of the faculty of psychology at the TUD Dresden University of Technology. In this system, students and individuals from the general population interested in being participants in psychological studies can register. 74 participants completed the experiment. 52 participants completed the informed consent but did not start to perform the experimental task or did not finish the task. We did not perform sample size calculations for our model-free analyses prior to data collection, as the combination of the task with the computational model was not previously used, and therefore, we could not estimate expected effect sizes from existing literature. We used parameter recovery and posterior predictive checks to validate our model-based analyses.

Four participants were excluded for lack of behavioral variability (they performed the same sequence of actions in more than 90% of all trials). The remaining 70 participants (50 women, 20 men) had a mean age of 24.1 years (SD = 4.6).

Participants were asked to self-report their gender as male, female, or diverse. All participants confirmed that they did not have dyschromatopsia. We did not collect data on race or ethnicity. Data was collected without interruption between 08.09.2021 and 10.11.2021.

Remuneration was a fixed amount of 10€ or class credit plus a performance-based bonus ($M$ = 2.58€, SD = 0.19€). The bonus was determined as a linear function from each participant's rewards acquired during the experiment, where a reward of 100 yielded 1ct. Participants were informed about the maximum of the bonus, but not the exact calculation.

The study was approved by the Institutional Review Board of the TUD Dresden University of Technology (protocol number EK 578122019) and conducted in accordance to ethical standards of the Declaration of Helsinki. All participants were informed about the purpose and the procedure of the study and gave written informed consent prior to the experiment. This study was not preregistered.

### Experimental task
Data collection was performed online. The task was built using lab.js[44] and hosted on the neurotests server of the TUD Dresden University of Technology, which is specifically designed for hosting lab.js tasks.

Participants had to navigate a Pacman-like character across a 5-by-5 grid using their keyboard to collect points matching a pre-defined trial-specific goal. In every trial, participants had to execute a sequence of four actions within a time limit of 6s (see Fig. 1). The action set was restricted to moves in three directions: diagonally to the upper left, diagonally to the upper right or directly downwards. This specific choice of navigation, inspired by the work of ref. 45, was designed to restrict the available sequences of actions participants could take. Exiting the grid's boundaries or revisiting a previously visited field was not possible. Any attempt to do so triggered a red warning message, requiring the participant to redo the move.

Upon each action, the character visually moved to the designated field and thereby collected the circle within that cell. Circles were colored to represent point values: Green circles represented positive points ranging from 10 to 60 in increments of 10, while red circles represented negative points ranging from −10 to −60. The shading of the color indicated the magnitude of points, with darker shades representing higher positive or negative values (see Fig. 2C). Additionally, Gaussian distributed noise (with $\mu = 0$, and $\sigma = 1.3$) was applied to the points earned from each move and the resulting value was rounded to the nearest integer. After each move, the points from the collected circle were displayed at the center of the grid, and the sum of points collected during that particular trial was displayed at the top left corner (see Fig. 1). The trial's total score was calculated as the sum of points from the combined sequence of four actions.

Importantly, the main goal of the task was to match the trial's points—achieved from the sequence of four actions—as closely as possible with a predefined, trial-specific goal. Participants' reward for each trial was then calculated based on the difference between the trial's total points and the

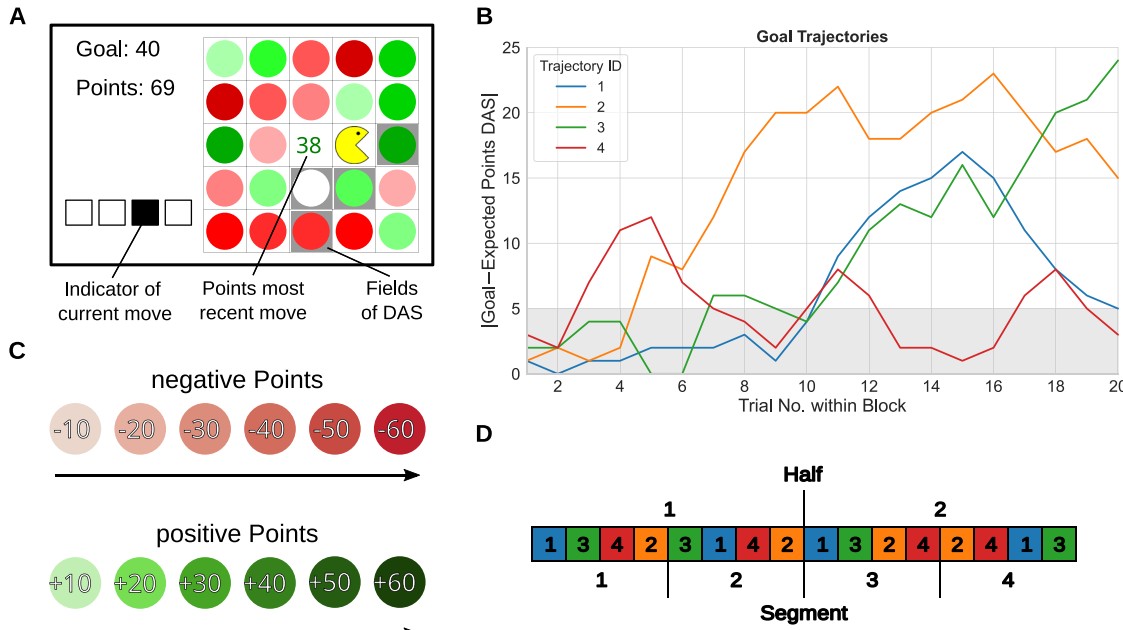

**Fig. 2 | Experimental task. A** Participants had to collect four colored circles by navigating a Pacman-style character on a 5-by-5 grid. Each circle's color indicated the number of points obtained when moving into the corresponding field (see main text for details). The points obtained for the most recent collected circle were displayed in the center of the grid (here 38). The point goal of the current trial was presented in the top-left corner. Below this goal, the current sum of points gathered up to the current move were displayed. Four squares located at the bottom left indicated the current move number. In the example, the participant was about to select its third move after having collected points from two circles. The default action sequence (DAS) fields were highlighted with a gray background. **B** Graph of the four goal point trajectories used. Each block consisted of 20 trials. The y-axis shows the absolute difference in points between the goal per trial, and the expected points one would obtain by using the DAS. The gray area represents trials where the DAS was one of the available action sequences with the highest expected reward. **C** Visual representation of points per color. **D** Order of goal trajectories within the experiment and mapping of blocks into halves and segments.

predefined goal. Smaller differences (in absolute value) led to higher rewards:

$$\text{Reward}_t = \max\{0, 100 - 2 \cdot |\text{Goal}_t - \text{Points}_t|\}. \tag{1}$$

The received reward was displayed as a green bar at the end of each trial in the feedback stage (see Fig. 1). If participants did not complete four moves within the time limit, no reward was earned and a red warning message appeared at the feedback stage on the left side. The sum of collected rewards defined the performance-based bonus payment.

Importantly, to ensure that participants repeat at least one sequence of actions frequently, and signal to participants that selecting a suboptimal action sequence is considered acceptable behavior, we introduced a default action sequence (DAS). This sequence was visually indicated by a gray background color of the four corresponding fields (see Fig. 2A) and participants were given information about the average number of points that could be collected with the DAS at the start of each trial at the center of the grid (see the left-most panel of Fig. 1). The fields, sequence of actions, and sum of collected points of the DAS were the same throughout the experiment. In the example of Fig. 1, the DAS comprised two downward moves first, followed by two consecutive up-right movements.

The experiment was divided into 16 blocks consisting of 20 trials each. The grid layout of points remained constant within a block, but changed between blocks. The points of the four circles of the DAS also differed across blocks but the sum of the points remained constant. The block sequence of the distributions of points were the same for each participant.

Within a block, goals changed over trials. We used four different trajectories of 20 goals each (see Fig. 2B). Goal trajectories differed by their maximum difference between the goal points and the expected points of the DAS, and their trend of this difference throughout the block. This difference ranged between 12 and 24 points. Note, that while the points that could be collected with the DAS stayed the same, the goals changed between trials, and therefore the rewards for the DAS varied.

Note that most studies that investigate the influence of repetition on value-based decisions use options with either fixed rewards (e.g., refs. 22,46), or fixed rewards combined with changing reward probabilities (e.g., refs. 29,47). In our task, through changing goals and grid layouts, we transparently vary the (relative) reward of the different options. With this, repeating an option throughout the task is not optimal. This results in a lower correlation between reward magnitude and behavioral repetition.

We selected these four trajectories to represent different principled trajectories that would make it difficult for participants to predict whether the DAS would remain optimal during the duration of a block. All four trajectories started out close to the expected points obtainable by the DAS so that the DAS was one of the best action sequences to select in the initial trials of a block. Only later into the block goals started to deviate from this initial value, or not. For example, one goal trajectory increased after half the block but then decreased again (trajectory ID 1, blue color) while another was remaining mostly close to points obtainable by the DAS (trajectory ID 4, red color).

With this procedure we effectively proposed an optimal sequence of actions at the first few trials of each block that was slowly devalued during subsequent trials. A repetition bias should manifest by increased DAS choices over the course of the experiment for the same expected rewards and should be detectable with summary statistics.

We subdivided the blocks into four segments of four blocks each (see Fig. 2D). Within each segment all four trajectories of goals were used once in a pseudo-random order so that no trajectory was repeated in two consecutive blocks. This order of goal trajectories was the same for each participant.

To promote the use of the DAS, we manipulated three features of the task. First, at least the first two trials of each block had a trial-specific goal close to the points of the DAS (see Fig. 2B). Overall, in 43.75% of all trials, the absolute difference between the trial goals and the expected points of the DAS was between zero and five points. Therefore, for these trials, due to the minimum difference of ten points between circles of different color, the DAS

was one of the available action sequences with the highest expected reward. In the remaining trials, there was always at least one sequence with a higher expected reward than the DAS.

Second, we only used a partial devaluation of the DAS: the lowest expected reward of the DAS was still about half of the maximum reward. This follows from the reward calculation (see Eq. (1)) and the maximum difference between goals and the expected DAS points of 24 (see Fig. 2B).

Third, using the DAS gave participants a probabilistic bonus reward of 20 during half of the blocks. These bonus blocks were distributed pseudo-randomly throughout the experiment to ensure that the bonus was available always during two of the four presentations of each goal trajectory. This probabilistic bonus could be earned for each trial within a bonus block, but only when using the DAS. The probability of receiving the bonus was $p = 0.25$, although the precise probability remained undisclosed to the participants. Participants were informed about upcoming bonus blocks right before they started. In addition, during the trial start phase, bonus trials were indicated by changing the color of DAS points from gray to blue. A blue bar next to the green reward bar during the feedback stage indicated the receipt of the bonus.

The experiment started with an elaborate training phase to ensure that participants understood the task. The first part involved an introduction to the navigation, which was followed by familiarizing participants with the color coding of the circles. Then, trial-specific goals and reward calculation was introduced. This was followed by introducing the DAS, and finally, the bonus factor was explained. During this part of the training participants had to meet no deadline and could spend as much time as they needed.

After this introductory phase, participants practiced two blocks with 20 trials each as they would appear later in the main experiment. One block was with probabilistic bonus and one without. The only difference to the main experiment was an extended deadline of 10s.

Between blocks, participants had the opportunity to take a self-determined break. The experiment, including training, had a total duration of ~60 min. The performance-dependent bonus rewards were determined by adding the rewards of all trials in the main experiment.

To ensure that participants understood the instructions of the task, we immediately asked them to describe their strategies and report any issues upon task completion. The majority of participants reported either no difficulties or only minor technical problems during a few trials. No participant indicated that they did not understand the task (all responses are available on OSF: https://osf.io/rh42a/). When asked about the strategy they used, most participants described meaningful strategies. For instance, 22 participants reported a strategy of using the DAS within a specified range of point difference between the goal and expected points obtained by using the DAS, and searching for alternative sequences outside this range. Additionally, we excluded participants with a lack of behavioral variability (i.e., proportion of DAS choices > 0.90), as we assumed that these participants did not fully understand the goal-directed nature of the task. All remaining participants achieved a higher average reward per trial (all > 64) than what would be achieved by responding randomly (50.43).

Data analysis was performed in Python using the packages NumPy, Pandas, ArviZ, Scipy's Stats module, scikit-posthocs, and pingouin. Plots were created with seaborn and matplotlib. Assumptions of $t$-tests and ANOVAS were checked and met, if not stated otherwise.

### Expected value with proxy and repetition bias model (EVPRM)

We made use of a previously published repetition-based learning model, the prior-based control model[42], for model-based data analyses. This model describes a mechanism for taking into account previous action sequences when making choices. The model counts how many times each action sequence $\pi$ has been chosen in the past. This contributes to the decision-making process as a prior distribution over policies $p(\pi)$ that represents the probability of selecting an action regardless of expected reward or any other task contingency: the influence of the prior over policies on action selection of a specific action increases depending on how many times this action has been chosen before. The model is complemented by a component $p(\hat{R}|\pi)$

based on expected rewards given the predicted outcomes of the performed actions (i.e., value-based). These two components play the role of priors and likelihood, respectively, to turn decision-making into a Bayesian inference process:

$$p(\pi|\hat{R}) \propto p(\hat{R}|\pi)\, p(\pi), \tag{2}$$

where $p(\pi|\hat{R})$ represents the posterior distribution that is defined by the probability of choosing policy $\pi$ given the reward structure $\hat{R}$; $p(\hat{R}|\pi)$ represents the expected reward $\hat{R}$ given policy $\pi$; and $p(\pi)$ is the prior over action sequences. The multiplication of the expected reward and the prior over policies balances the influences of goal-directed planning and past behavior on action selection. Intuitively, the goal-directed component $p(\hat{R}|\pi)$ represents the value-based part of making a decision, i.e., a participant simply selects the action that gives the highest expected reward, while the prior over action sequences $p(\pi)$ implements the repetition bias.

In our experimental task, the reward structure $\hat{R}$, i.e., the expected reward for each action sequence $\pi$, can in principle be calculated given the information available to participants: the points for each one of the squares on the grid is shown on the screen, so participants could calculate the points of every of the possible 36 sequence of actions $\pi$ and determine the expected reward with the exception of a noise term that is not influenced by $\pi$. However, as there is a deadline of six seconds, the calculation becomes unfeasible. To account for this, we posit that participants rely on prior beliefs or approximations they might have acquired in previous trials.

For the proposed expected value with proxy and repetition bias model (EVPRM), we assumed a reward structure $\hat{R}$ that depends on past observations made by the participant: for sequences that have been already observed during the current block, the model uses the exact observed reward; for the unobserved sequences, it uses an approximated reward $R_0$, which we assumed participants approximate based on their experiences during previous blocks and training. The approximated reward $R_0$ is a free parameter and indicates the individual expected reward for all yet unobserved sequences of actions. As an exception, the DAS was always assumed to be an observed sequence of actions because the points of the DAS were communicated at the initial phase of each trial. In addition, the expected reward of the DAS included the probabilistic bonus reward. Therefore, in all bonus trials, the expected reward of the DAS was enhanced by 5 (probabilistic bonus of 20 with a probability of 0.25). With this, the reward structure in the EVPRM is as follows:

$$p(\hat{R}|\pi) = \left( \frac{\hat{R}_\pi}{\sum_{\pi \in \lambda} \hat{R}_\pi} \right)^\beta \tag{3}$$

$$\hat{R}_{\pi_t} = \begin{cases} R_{\pi_t} & \text{if } \pi \in \{\tilde{\pi}_{1:t}\} \\ R_0 & \text{otherwise} \end{cases}, \tag{4}$$

where $\pi = \{(a_1, a_2, a_3, a_4)|a_i \in \{\searrow, \downarrow, \nearrow\}\}$, with $a_i \in \{\searrow, \downarrow, \nearrow\}$ represents the three movement directions, $\hat{R}_\pi$ is the expected reward of the sequence of actions $\pi$, $\sum_{\pi \in \lambda} \hat{R}_\pi$ is the sum of expected rewards of all sequences of actions, with $\lambda$ representing all possible action sequences, $\beta$ is a free parameter representing the precision over expected rewards, $R_\pi$ is the expected reward for the sequence of actions $\pi$, $R_0$ is the approximated reward for unobserved sequences of actions, $\tilde{\pi}_{1:t} = \{\tilde{\pi}_1, \tilde{\pi}_2, \ldots, \tilde{\pi}_t\}$ are the performed sequences of actions up to trial $t$. Note that we chose $p(\hat{R}|\pi)$ as a fraction to stay close to the Bayesian framework in ref. 42 and to have a comparable equation to the prior below.

The free parameter $\beta$ represents the precision over expected rewards: values of $\beta > 1$ leads to more concentrated probabilities that favor the choice of the sequences of actions with the highest expected rewards and values of $\beta < 1$ lead to more uniformly distributed probabilities, enabling greater exploration of different choices.

The prior over action sequences $p(\pi)$ was defined, as by ref. 42, by a counter $\gamma$ for the number of times the respective sequence of actions has

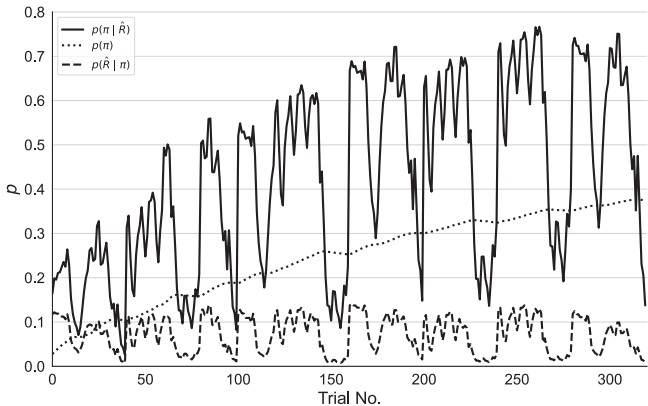

**Fig. 3 | Simulation of task with expected value with proxy and repetition bias model (EVPRM).** Means for probability of selecting one specific sequence of actions $\pi$, $p(\pi|\hat{R})$ (solid line), the prior over policies for $\pi$, $p(\pi)$ (dotted line), and the expected reward for $\pi$, $p(\hat{R}|\pi)$ (dashed line) over $N = 100$ simulations with the following parameters: precision over expected reward $\beta = 5$, approximated reward $R_0 = 70$, and repetition bias strength $h = 0.91$. While expected rewards are in a constant range throughout the task, the prior over policies increases and accordingly the choice probability.

been used in the past, and the initial count $\alpha_{\text{init}}$ is a free parameter that was equal for each sequence of actions:

$$p(\pi) = \frac{\alpha_\pi}{\sum_{\pi \in \lambda} \alpha_\pi} \qquad (5)$$

$$\alpha_{\pi_t} = \alpha_{\text{init}} + \gamma_{\pi_t} \qquad (6)$$

$$(\gamma_{1:t})_i = \sum_{\tau=1}^{t} \delta_{i\tau} \qquad (7)$$

$$\delta_{i\tau} = \begin{cases} 1 & \text{if } \pi_i \text{was used at } t \\ 0 & \text{otherwise} \end{cases}, \qquad (8)$$

where $\alpha_{\pi_t}$ is the repetition bias strength at trial $t$, $\alpha_{\text{init}}$ is the initial count, $\gamma_{\pi_t}$ is the counter of how many times the sequence of actions $\pi$ was performed until trial $t$, $\delta_{i\tau}$ is the Kronecker delta.

Following ref. 42, the free parameter initial count $\alpha_{\text{init}}$ influences the strength of the repetition bias. A low initial count, e.g., $\alpha_{\text{init}} = 1$, leads to a strong repetition bias. As $\alpha_{\text{init}}$ defines the counter for all sequences of actions, the increase of $\gamma$ by 1, after a sequence of actions was performed, leads to a substantial increase of the prior over policies for this sequence. In contrast, a high initial count, e.g., $\alpha_{\text{init}} = 100$, leads to a weak increase of the prior over policies if a sequence of actions is performed.

Finally, our model can make decisions at every trial by sampling from the categorical posterior probability distribution over possible $\pi$, defined as: $p(\pi|\hat{R}, \alpha_\pi)$, which is the probability of sampling each sequence of actions $\pi$, at each trial depending on the assumed reward structure $\hat{R}$ and the prior over policies $\alpha_\pi$:

$$p(\pi|\hat{R}, \alpha_\pi) \propto p(\hat{R}|\pi)p(\pi) \qquad (9)$$

$$p(\pi|\hat{R}, \alpha_\pi) \propto \left(\frac{\hat{R}_\pi}{\sum_{\pi \in \lambda} \hat{R}_\pi}\right)^\beta \cdot \frac{\alpha_\pi}{\sum_{\pi \in \lambda} \alpha_\pi} \qquad (10)$$

By changing the free parameters, we can change the behavior of the agent: At one end, with a high initial count $\alpha_{\text{init}}$, an agent will be minimally influenced by its past behavior and is nearly completely goal-directed. At the other end, with a low initial count $\alpha_{\text{init}}$, agent behavior is less influenced by

expected rewards and thus has a strong repetition bias of past action sequences. In addition, a precision over expected rewards $\beta$ close to 0 represents the case in which the agent is uncertain about the learned reward structure and will tend to choose behavior based on the repetition bias.

In Fig. 3 we simulate an experimental session with our model, focusing on one action sequence $\pi$, the DAS. In the simulations, the model has a high influence of past behavior ($\alpha_{\text{init}} = 1.1$). The used precision over expected rewards ($\beta = 5$) moderately pronounced the distribution of expected rewards. Based on the changing goals the expected reward $p(\hat{R}|\pi)$ for this action sequence changes in a constant range from trial to trial throughout the experiment. But the prior over policies $p(\pi)$ for this action sequence increases slowly over time, because it is performed repeatedly. One can see that in trials where the expected reward is relatively high, the resulting posterior $p(\pi|\hat{R})$ is high as well. This means the resulting choice probability is driven by expected rewards, represented by the first term on the right-hand side of Eq. (10). In addition, as the prior is slowly increasing, there is a growing contribution of the repetition bias, given by the second term of Eq. (10). Hence, the repetition bias increases the choice probability but actions are in principle still modulated by expected rewards. Effectively, in the example, the repetition bias increases the choice frequency from roughly 0.3 in the first 50 trials to roughly 0.7 in the last 50 trials, when there is a relatively large expected reward.

Note that the original model by ref. 42 was formulated within a planning as inference[48] and active inference[49,50] framework to calculate the posterior distributions for action selection. We adapted the key idea: the posterior probabilities are based on the product of a function over the expected rewards and the prior over policies. Here, for our purposes, we simplified this model to derive a relatively straightforward observation model so that we could use Bayesian inference for fitting the model's free parameters to participant data. Furthermore, the model calculates probabilities based on past and current observations and does not use any kind of future forward planning. It is therefore related to RL models, which also calculate subjective values for the possible actions based on the current expected rewards and the reward history[29,41,51].

To ensure that parameter inference works well for a meaningful range of parameters, we performed extensive parameter recovery studies for the EVPRM and all alternative models (for details see Supplementary Fig. 4). Furthermore, to assess the influence of prior information on posterior inference, we calculated the shrinkage between prior and posterior based on prior and posterior standard deviations[52] (see Supplementary Fig. 5). A strong influence of data on the posterior is indicated by high shrinkage (i.e., values approaching 1). We found most shrinkage values above 0.4, and for our main model parameter, the repetition bias strength $h$, half of the participants ($n = 35$) showed a high shrinkage above 0.8. Therefore we concluded that our posteriors are mainly defined by the data and not considerably affected by our prior distributions.

## Alternative models

The proposed EVPRM makes two assumptions: (1) the repetition bias influences action selection, and (2) participants used an approximated reward for unobserved sequences of actions. To test against plausible alternative explanations, we formulated three additional models. These models differ in their assumed reward structure $\hat{R}$ and as a critical distinction, they do not include the prior over policies $p(\pi)$. In what follows, we describe these three alternative models.

## Expected value with proxy and default bias model (EVPBM).

An alternative explanation for the repetition of the DAS would be a bias specifically for the DAS but not a general repetition bias, also for other sequences than the DAS, as formulated in the EVPRM. To implement this assumption, we derived a model variant that had an exclusive and constant bias for the DAS. In other words, this model assumes that during training participants developed a bias for choosing the DAS but did not have a slowly increasing repetition bias or a preference for repeating other action sequences. The difference to the EVPRM is that a constant bias for

the DAS would not be dependent on past behavior in the main experiment.

To model this bias, we added a constant term as a free model parameter to the expected reward of the DAS:

$$p(\pi|\hat{R}) \propto \left(\frac{\hat{R}_\pi}{\sum_{\pi \in \lambda} \hat{R}_\pi}\right)^\beta \qquad (11)$$

$$\hat{R}_{\pi_t} = \begin{cases} R_{DAS_t} + b_{DAS} & \text{if } \pi = DAS \\ R_{\pi_t} & \text{if } \pi \in \{\tilde{\pi}_{1:t}\}, \\ R_0 & \text{otherwise} \end{cases} \qquad (12)$$

where $\pi$ is a sequence of four actions defined as above, $\hat{R}$ is the assumed reward structure, $\hat{R}_\pi$ is the expected reward for the sequence of actions $\pi$, $\beta$ is the free model parameter representing precision over expected rewards, $R_0$ the free model parameter of approximated rewards for yet unobserved sequences of actions, $\tilde{\pi}_{1:t} = \{\tilde{\pi}_1, \tilde{\pi}_2, \ldots, \tilde{\pi}_t\}$ are the performed sequences of actions up to trial $t$, and $b_{DAS}$ is the bias for $\pi_{DAS}$.

**Expected value with proxy model (EVPM).** A second alternative explanation of the choice data is that repetition does not influence action selection at all. Therefore, contrary to the EVPRM, participants' behavior is not affected by past behavior, but determined by expected rewards only. To implement this assumption, we removed the prior over policies from the EVPRM to have a model that is solely dependent on the expected reward structure:

$$p(\pi|\hat{R}) \propto \left(\frac{\hat{R}_\pi}{\sum_{\pi \in \lambda} \hat{R}_\pi}\right)^\beta \qquad (13)$$

$$\hat{R}_{\pi_t} = \begin{cases} R_{\pi_t} & \text{if } \pi \in \{\tilde{\pi}_{1:t}\} \\ R_0 & \text{otherwise} \end{cases}, \qquad (14)$$

where $\pi$ is a sequence of actions defined as before, $\hat{R}$ is the assumed reward structure, $\tilde{\pi}_{1:t} = \{\tilde{\pi}_1, \tilde{\pi}_2, \ldots, \tilde{\pi}_t\}$ are the performed sequences of actions up to trial $t$, $\beta$ is the precision over expected rewards and a free model parameter, $R_\pi$ the expected reward for a sequence of actions $\pi$, and $R_0$ the free model parameter of approximated reward for unobserved sequences.

**Expected value model (EVM).** The EVPM relies on the approximated reward for unobserved sequences of actions $R_0$. An alternative is that participants indeed were able to calculate expected rewards for all sequences of actions. To implement this assumption we instantiated a model without the approximated reward parameter and used expected reward $R_\pi$ instead. Therefore, contrary to the other candidate models, this model performs action selection solely driven by reward seeking with exact expected rewards. We implemented this as:

$$p(\pi|R) \propto \left(\frac{R_\pi}{\sum_{\pi \in \lambda} R_\pi}\right)^\beta, \qquad (15)$$

where $\pi$ is a sequence of four actions defined as before, $R_\pi$ is the expected reward of the sequence of actions $\pi$, and $\beta$ is the free model parameter representing precision over expected rewards.

**Model fitting**
Parameter estimation was done in Python with PyMC[53][version 5.0.1]. using the No U-Turn Sampler[54]. We obtained 4000 samples from four chains of length 1000 (1000 warm-up samples).

We used the following weakly informative prior distributions for the free model parameters: $\beta \sim Gamma(3, 1)$, $R_0 \sim Gamma(55, 0.75)$, $h = \frac{1}{\alpha_{init}} \sim Beta(3, 3)$, and $b_{DAS} \sim Gamma(3, 0.1)$. We used the same priors for all candidate models.

**Model comparison**
Model comparison was based on using leave-one-out cross-validation approximated by Pareto-smoothed importance sampling (PSIS-LOO)[55]. This information criterion calculates the pointwise out-of-sample predictive accuracy from a fitted Bayesian model. Crucially, it penalizes models with more parameters. We calculated the expected log point-wise predictive density (elpd) and the corresponding standard error (SE) on the deviance scale ($-2elpd$). Lower values of PSIS-LOO indicate higher predictive accuracy. Calculation of PSIS-LOO scores was performed with ArviZ [56][version 0.7.0].

**Parameter distributions of EVPRM**
To better compare individual repetition bias strengths we used the inverse of the initial count $\alpha_{init}$: $h = \frac{1}{\alpha_{init}}$ (see Eq. (6)). This repetition bias strength $h$ has a value range from 0 to 1, where values near 1 indicated a strong repetition bias and values around 0 indicate a weak repetition bias.

Across participants, repetition bias strength varied from very low values between close to 0 and 0.2 to medium to strong values between 0.5 and 0.9 (see Supplementary Fig. 2A). The inferred $\beta$ values (the precision over expected reward) spread between values very close to 0 and high values up to 16 (see Supplementary Fig. 2B). The approximated reward $R_0$ for unobserved sequences of actions showed a broad range of values between around 10 and around 70 (see Supplementary Fig. 2C). For summary statistics of all posterior distributions for all parameters see Supplementary Table 3.

**Posterior predictive checks for EVPRM**
We conducted posterior predictive checks[57] to assess if the fitted EVPRM can replicate the behavior of the participants. We used PyMC to simulate choices of 1000 agents for each participant based on the model and posterior. The parameters of the agents were drawn from the posterior distributions.

We calculated the proportion of correctly predicted choices for each participant over all agents. These proportions of correctly predicted choices showed a wide range from 4.9% to 86.3%, but all proportion were above the chance level of 2.7% (see Supplementary Fig. 3A). On the group level the EVPRM predicted DAS choices better (74.9%) compared to non-DAS choices (19.1%, chance level = 2, 8%). We further calculated correlations between the proportions of matched choices and the posterior means of the inferred parameters and the proportion of DAS choices $p(DAS)$. Here, EVPRM predicted choices of participants with (1) weak repetition bias strength $h$ better than participants with strong repetition bias strength, $r = 0.55$, $p < 0.001$, CI = [0.36, 0.70], (2) participants with higher precision over expected rewards $\beta$ better than participants with lower precision over expected rewards, $r = 0.73$, $p < 0.001$, CI = [0.60, 0.82], and (3) participants with higher proportions of $p(DAS)$ better than participants with lower proportions of $p(DAS)$, $r = 0.84$, $p < 0.001$, CI = [0.76, 0.90] (see Supplementary Fig. 3B). The correlation with the approximated reward $R_0$ was not significant, $r = 0.12$, $p = 0.248$, CI = [0.11, $-0.35$].

**Reporting summary**
Further information on research design is available in the Nature Portfolio Reporting Summary linked to this article.

**Results**
We designed a sequential decision-making task, the Y-navigation task (Y-NAT), to show the repetition bias (see "Methods"). For this grid-world task, participants had to collect points with four moves within a time limit of 6s and match a trial-specific goal sum of points as closely as possible (see Fig. 1). Participants were required to complete 16 blocks, each comprising 20 trials, resulting in a total of 320 trials.

In order to ensure the frequent repetition of at least one sequence of actions and implicitly signal to highly motivated participants that repeating a potentially suboptimal sequence, as opposed to always trying to find the best sequence, is acceptable task behavior, a default action sequence (DAS)

**Table 1 | Descriptive statistics of performance measures for all trials and halves of the experiment**

|  | All Trials | | First Half | | Second Half | | t-Test | | | |
|---|---|---|---|---|---|---|---|---|---|---|
|  | **M** | **SD** | **M** | **SD** | **M** | **SD** | **t** | **p** | **d** | **CI** |
| p(DAS) | 0.54 | 0.19 | 0.53 | 0.20 | 0.55 | 0.19 | −1.62 | 0.055 | 0.10 | [−∞, 0.00] |
| Reward | 80.69 | 5.99 | 79.58 | 6.66 | 81.80 | 5.96 | −4.58 | <0.001 | 0.35 | [−∞, -1.41] |
| RT (ms) | 1677.09 | 398.72 | 1820.84 | 450.34 | 1535.60 | 390.85 | 8.76 | <0.001 | 0.68 | [230.97, ∞] |
| Time Outs | 4.74 | 3.70 | 3.11 | 2.56 | 1.63 | 1.88 | 4.86 | <0.001 | 0.66 | [0.01, ∞] |

Means over all participants of p(DAS), reward per trial, reaction times and the number of time outs, separately for all trials, and the first and second halves of the experiment. The one-sided t-tests tested for significant differences between first and second half. For p(DAS) and reward we tested if the mean of the first half is smaller than the mean of the second half. For RT and time outs, we tested if the means of the first half are greater than the means of the second half. N = 70 participants.

p(DAS) proportion of DAS choices, reward mean reward per trial, RT reaction time, DAS default action sequence, M mean, SD standard deviation, t t-statistic, p p-value, d Cohen's d, CI 95% confidence interval of mean difference.

was highlighted. Using the DAS resulted in the highest expected reward in less than half of the trials (43.75%), with the lowest expected reward of the DAS being about half of the maximum reward. Furthermore, in half of the blocks, a probabilistic bonus could be earned when using the DAS.

In what follows we first present the results from standard behavioral analyses based on summary statistics and then move on to a model-based analysis.

**Behavioral analysis**

Our first approach was to find evidence of a repetition bias using inference statistics based on summary measures. For our task, we expected that a repetition bias manifests in the following ways: (1) an increase, over the course of the experiment, in the usage of the most frequently used sequence of actions; (2) an increase, over the course of the experiment, in the selection of the most frequently used sequence of actions in trials where this sequence of actions did not have the highest expected reward; (3) an increase, over the course of the experiment, to perform parts of the most frequently used sequence of actions, and (4) a decrease, over the course of the experiment, in the number of different sequences of actions being used.

We determined the proportion of trials in which the DAS was executed, p(DAS), for each participant. As expected, the DAS was used in more than half of the trials (p(DAS) = 0.54, SD = 0.19), and 66 participants (94%) used the proposed DAS most frequently (see Table 1). We found the expected difference in the proportion of DAS choices between the bonus (p(DAS) = 0.57, SD = 0.19) and the no bonus condition (p(DAS) = 0.52, SD = 0.19), $t(69) = −4.68, p < 0.001, d = 0.26$, CI = [−∞, −0.03] (see Supplementary Table 1 for all descriptive statistics depending on the bonus condition).

Furthermore, post-hoc comparisons to test for gender differences were conducted. Women and men did not differ significantly in their propensity to use the DAS (women: $M = 0.53$, SD = 0.19; men: $M = 0.57$, SD = 0.17; Welch-t-test, $t(39.54) = −0.81$, $p = 0.421$, $d = 0.20$, CI = [−0.13, 0.06]). Therefore, we could not find gender-related differences in the propensity to use the proposed DAS. However, men achieved a significant higher mean reward per trial than women (women: $M = 79.89$, SD = 5.93; men: $M = 82.69$, SD = 5.79; Mann-Whitney $U$ test, because the assumption of normality was violated, $U = 302.00, p = 0.010$, CLES = 0.30, CI = [0.16, 0.46]). Furthermore, men were significantly faster than women (women: $M = 1755.47$ms, SD = 415.56ms; men: $M = 1481.13$ms, SD = 274.59ms; Mann-Whitney $U$ test, because the assumption of normality was violated, $U = 723.00$, $p = 0.004$, CLES = 0.72, CI = [0.59, 0.84]), and men had significantly fewer time outs (women: $M = 5.32$, SD = 3.89; men: $M = 3.30$, SD = 2.74; Mann-Whitney $U$ test, because the assumption of normality was violated, $U = 654.50, p = 0.044$, CLES = 0.65, CI = [0.52, 0.78]). Of these results, only the differences in the proportion of DAS choices would have direct relevance to our results. As this difference was non-significant, we do not further analyze the differences in the other performance measures.

Here, in our standard analysis, we focused most of the subsequent analyses on the DAS because participants used the DAS more frequently than expected when considering expected rewards, and the DAS was used

most frequently by nearly all of the participants. We conducted four statistical analyses to test for a repetition bias: we tested for an increased usage of the DAS over time, an increased usage of the DAS even when the sequence did not yield the highest expected reward over time, and an increased usage of parts of the DAS, when the full sequence was not performed. Furthermore, we tested for a decrease of behavioral variability as an indicator of a repetition bias. We describe the results of these analyses in the following sections.

**Increase of DAS usage.** We assessed whether there was an increase in DAS usage over the course of the experiment. We repeated trials with the same differences between the trial-specific goals and the expected points of the DAS across halves and four segments (see Fig. 2D), and consequently expected rewards for the DAS were equivalent across the halves and the segments. The expected points for the DAS were communicated at the beginning of each trial and participants were able to calculate the expected reward for the DAS. Therefore, participants' proportion of DAS choices should not change over halves and segments if they were guided only by expected rewards. However, if a repetition bias influenced participants' choices, DAS usage should have increased with time.

We compared the average proportion of DAS choices of all participants between the first and second half of the experiment, and over the four segments (comprising four subsequent blocks, see also Fig. 2D in "Methods"). During the first half, over all participants, the DAS was used in 53.3% (SD = 19.7%) of the trials, whereas in the second half the DAS was used in 55.2% (SD = 18.8%) of the trials (see Fig. 4A). A one-sided t-test for related samples based on the individual differences of all participants only showed a non-significant difference, $t(69) = −1.62, p = 0.055, d = 0.10$, CI = [−∞, 0.00]. Similarly, there was only a non-significant increase of DAS usage across the four segments, repeated measures ANOVA with $F(3, 207) = 1.52$, $p = 0.21, \eta_G^2 = 0.003$, CI = [0.001, 0.007], see Fig. 4A.

**Increase in DAS usage in trials where DAS is not optimal.** Although we did not find a significant increase in DAS usage over the course of the experiment, a repetition bias for the DAS should increase the probability of selecting the DAS irrespective of the expected reward of the DAS. We expected this because at the beginning of each block the DAS was an optimal choice (see "Methods" and Fig. 2B) participants were incentivized to use the DAS in the first trials of each block. This incentivized repetition of the DAS would bias participants towards choosing the DAS even when it did not yield the highest expected reward during later trials of the block. However, if goals shift into the range where the DAS is one of the optimal sequences (e.g., trials 7–10 of goal trajectory 4, see Fig. 2B), a strong repetition bias for specific sequences can decrease the probability that participants switch back to the again-optimal DAS. These two opposing effects together could explain why we found no significant overall increase in DAS usage so far.

To assess if participants changed their DAS usage depending on expected reward, we split up trials based on whether the DAS was one of the available sequences of actions with the highest expected reward, or not. We

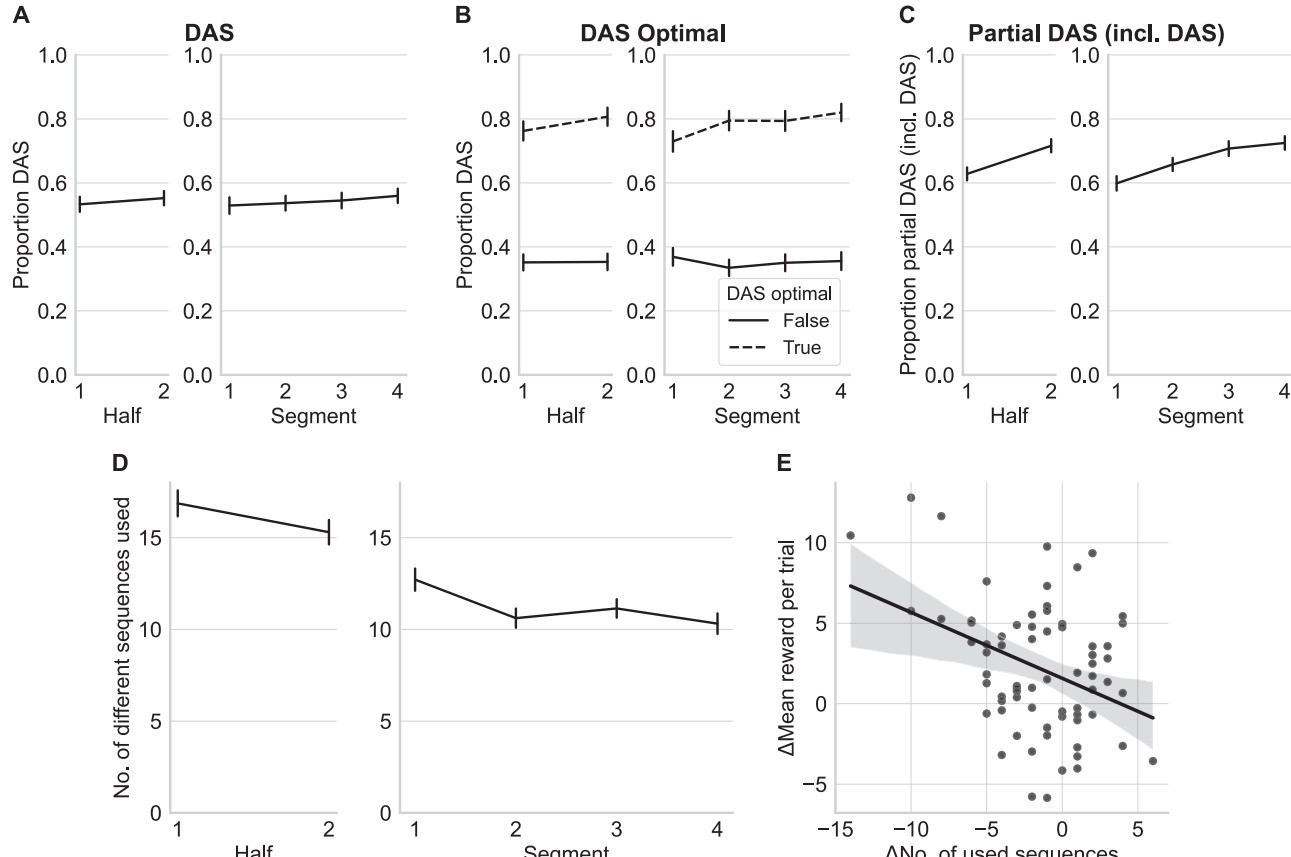

**Fig. 4 | Behavioral analysis.** **A** Mean proportional use of default action sequence (DAS) over the two halves (early/late) and four segments of the experiment. **B** Mean proportional use of DAS depending on if the DAS was one of the sequences with the highest expected reward (optimal), separated by halves and by the four segments. **C** Mean proportional use of partial DAS (including DAS) trials depending on halves and the four segments of the experiment. **D** Mean number of used different sequences of actions over the two halves (early/late) and four segments of the experiment. Black lines represent means over all participants. Error bars represent standard errors (*SE*). **E** Correlation between the difference of used sequences and the difference of mean reward per trial between the first and the second half. Each point represents one participant. The thick solid line represents linear regression model fitted to the data. Gray area represents 95% confidence interval for the regression. *N* = 70 participants.

determined a use of DAS as optimal if the absolute difference between the points obtained by the DAS and the goal was ≤5 points, because the difference between two adjacent colors was 10 points (see "Methods"). We conducted a repeated measures ANOVA with the proportion of DAS choices as dependent variable and the factors (1) halves of the experiment, and (2) DAS optimality. The main effect of expected reward was significant, $F(1, 69) = 293.89, p < 0.001, \eta_G^2 = 0.474, \text{CI} = [0.449, 0.483]$. The main effect experimental halves was not significant, $F(1, 69) = 3.88, p = 0.053, \eta_G^2 = 0.003, \text{CI} = [0.001, 0.005]$ (see Fig. 4B). We repeated this analysis with four segments instead of halves of the experiment as a factor. Again, the main effect of expected reward was significant, $F(1, 69) = 239.15, p < 0.001, \eta_G^2 = 0.451, \text{CI} = [0.420, 0.454]$ and the main effect of segment was not significant, $F(3, 207) = 2.36, p = 0.077, \eta_G^2 = 0.003, \text{CI} = [0.001, 0.006]$ (see Fig. 4B).

Taken together, using summary measures, we did not find evidence for an increase in DAS use for trials where the DAS was not one of the sequences of actions with the highest expected reward.

**Increase in DAS parts.** As participants had to execute a sequence of four moves in each trial, a repetition bias effect may have expressed itself by an increase of the probability of repeating at least the first move(s) of a sequence of actions. Due to the small trial-wise changes of the goal points (see Fig. 2B), a possible strategy for participants would be to repeat the first move(s) of a sequence of actions but deviate from this sequence after these initial move(s), depending on the goal points.

We categorized the used sequences of actions into three categories: a DAS trial (when the full DAS was executed), a partial DAS trial (trial with a deviation from the DAS), or a DAS-independent trial (a completely different sequence). Trials that were categorized as partial DAS trials were defined by selecting at least the first move in accordance with the DAS, but not using the complete DAS.

To test for an increase in repeating the first move(s) of the DAS, we tested for differences in the proportion of partial DAS trials (including DAS trials) over halves and segments. A one-sided *t*-test for related samples revealed that the proportion of partial DAS (including DAS) trials significantly increased from the first half of the experiment to the second half, $t(69) = -6.46, p < 0.001, d = 0.49, \text{CI} = [\infty, -0.06]$ (see Fig. 4C). Similarly, a repeated measures ANOVA over segments showed a significant increase of partial DAS (including DAS) use over time, $F(3, 207) = 19.53, p < 0.001, \eta_G^2 = 0.06, \text{CI} = [0.04, 0.07]$ (see Fig. 4C).

**Decrease in behavioral variability.** The repetition bias effect might also lead to an increasing (over time) probability of repeating performed sequences of actions other than the DAS. This would lead to a lower number of different action sequences being performed in later stages of the experiment. To test this, we analyzed the number of used different sequences of actions between the halves and the four segments of the experiment. The mean number of used sequences of actions showed a small significant decrease from the first (16.87, SD = 5.94) to the second half (15.30, SD = 5.58), $t(69) = 3.53, p < 0.001, d = 0.27, \text{CI} = [0.83, \infty]$

(see Fig. 4D). A repeated measures ANOVA with segments as factor showed a significant effect as well, $F(3, 207) = 13.88, p < 0.001, \eta_G^2 = 0.04$, CI = [0.02, 0.06] (see Fig. 4D). Here the number of used sequences decreased significantly from the first to the second segment, but was stable throughout the following segments.

A decrease in behavioral variability could also reflect learning. Maybe participants selected some sequences of actions with very low rewards once during the first half of the experiment, but learned how to avoid selecting sequences with low rewards. To assess if this decrease in behavioral variability was caused by learning to find rewarding sequences more easily, we calculated the correlation of (1) the difference in the number of used sequences between the first and second half of the experiment with (2) the difference of the mean reward per trial between the first and the second half. A negative correlation would indicate that participants who performed fewer sequences improved their ability to find rewarding sequences. This correlation showed a significant relationship in the expected direction, $r = -0.38, p = 0.001$, CI = [-0.56, -0.15] (see Fig. 4E), but could have been driven by few participants with high increases in reward in combination with a high decrease in used sequences. Despite this potential limitation, our analysis suggests that a decrease in behavioral variability was mainly associated with an increase in mean rewards, as more participants with a decrease in behavioral variability improved their mean rewards. This in turn suggests that the decrease in behavioral variability could be caused by learning to identify sequences with high rewards, over the course of the experiment.

In summary, we found only weak evidence for a repetition bias effect, using a summary statistics approach. Participants used the DAS more frequently than expected, but we only found hints for a repetition bias when also considering partial DAS choices, or by analyzing the change in the number of used sequences. Conducting a similar analysis on the second most used sequence of actions or a Chi-square test of independence with all sequences of actions would not be meaningful: In contrast to the DAS, the expected rewards for all other sequences of actions differed between the first and second half of the experiment due to the grid layout changes between blocks. Hence, a sequence of actions that was frequently used during the first half of the experiment may generate very low rewards in the second half. As the repetition bias only increases the probability to repeat actions, we expect that action selection is mostly still guided by expected rewards, and participants should prefer to switch to action sequences with higher rewards[58,59]. Therefore, to test for a repetition bias, the analysis should also take into account expected rewards. For this reason, we next turn to a model-based analysis, which enables us to simultaneously consider both the repetition of action sequences and reward seeking as effects on observed choices.

**Model-based analysis**

A potential issue with our analyses above is the limited focus on behavioral measures for one specific sequence of actions, for example how many times the DAS was used in the first and second half of the experiment, thereby not considering expected rewards and other action sequences.

To consider all sequences of actions, and expected reward and repetition bias simultaneously, we used an adapted version of the prior-based control model[42], which we call here the "expected value with proxy reward and repetition bias model" (EVPRM). For the full model specification and details, see "Methods" and Fig. 3.

This model calculates the probability of selecting an action based on the balance of two components: the repetition bias mechanism, and expected reward. Crucially, the influence of the repetition bias is modeled by summing the number of times each action sequence has been used in the past $\gamma_\pi$. This component is weighted by a free model parameter $\alpha_{init}$ that determines the repetition bias strength (see Expected value with proxy and repetition bias model (EVPRM) and Eq. (5)). Using this model, the focus is not restricted to one sequence of actions and the repetition bias strength quantifies the influence of past behavior on action selection for all possible sequences of actions.

Moreover, the EVPRM incorporates the influence of expected rewards of all sequences of actions on action selection. Like past behavior, expected rewards are weighted by a free parameter $\beta$ that quantifies the individual precision on normalized expected rewards (see Eq. (3)). A high precision $\beta$ leads to pronounced probabilities and a stronger influence of expected rewards on action selection, while a low precision leads to more uniformly distributed probabilities and lower influence of expected rewards on action selection.

As the influence of expected reward and past behavior is modeled by two different parameters, $\alpha_{init}$ and $\beta$, we can disambiguate between effects on behavior by a low precision on expected rewards and effects on behavior driven by a strong repetition bias. Crucially, as we will show below this makes it possible to explain behavior that is both influenced by the current expected rewards and by past behavior.

To test whether the repetition bias effect is required at all to explain the behavioral data, we considered three alternative models that do not include an explicit repetition term. First, we used the expected value model (EVM), which posits that participants know the exact expected rewards and performed actions to solely maximize the expected rewards. However, as explained in the "Methods" section, this model would require infeasible computations made by participants as they perform the task. Second, we used a model that is based on expected reward structure only. For this expected value with proxy model (EVPM), the reward is known for those sequences that have been chosen before, but for all others an approximated value $R_0$ is used, which we assume participants estimated based on task instructions and training. Third, we considered the possibility that participants prefer the DAS based on the initial training and instructions. To model this, we used an extension of the EVPM, the expected value with proxy and default bias model (EVPBM), which has a constant bias in favor of the DAS to account for the observed high probability of DAS choices in our data. For details on the models, see "Methods".

**Model comparison.** We calculated the predictive accuracy of the four cognitive models (EVPRM, EVM, EVPM, EVPBM) at the group level. We used the leave-one-out information criterion (LOOIC) [55] that evaluates model fit but also penalizes for model complexity (see "Methods" for details). Lower LOOIC values indicate a higher predictive accuracy, i.e., a lower difference between model predictions and observed data. We found that the EVPRM, the model including the postulated repetition bias effect, showed the highest predictive accuracy (LOOIC = 69,942.82, SE = 825.94) compared to the EVPBM (LOOIC = 73,285.86, SE = 983.73), the EVPM (LOOIC = 74,301.96, SE = 1018.95), and the EVM (LOOIC = 162,308.58, SE = 1355.02) (see Table 2). Because of its low predictive accuracy we excluded the EVM from further analyses. Also, a random response model failed in explaining the data (LOOIC = 158,162.19, SE = 0.00) and was not further considered.

Following the guidelines from ref. 60 for interpreting LOOIC values, we found that the EVPRM described the data significantly better than the second-best model EVPBM: the standard error of the LOOIC differences $dSE$ between EVPRM and EVPBM was substantially smaller than the difference in LOOIC between these models dLOOIC (see Table 2). To assess how well the models explained behavior at the participant level, we compared the LOOICs of the three remaining candidate models for each participant individually.

First, we counted how many participants were fitted best by each of the three candidate models. This classification showed no clear pattern, as a considerable number of participants were equally well explained by each of the models (see Table 2). Although EVPRM was the best model on the group level, the behavior of only 27 out of 70 participants (ca. 39%) was described best by EVPRM.

As a next step, we looked at the individual LOOIC values of all models (see Fig. 5). Here, most of the participants whose behavior was described best by EVPRM showed a difference between the LOOICs of the candidate models, indicating that the EVPRM explained behavior better than the alternative models. In contrast, the LOOICs of those participants whose

## Table 2 | Results of model comparison

| Model | LOOIC | SE | pLOOIC | dLOOIC | dSE | % of participants with best fit (*n*) |
|---|---|---|---|---|---|---|
| EVPRM | 69,942.82 | 825.94 | 810.92 | 0.00 | 0.00 | 38.6% (27) |
| EVPBM | 73,285.86 | 983.73 | 1054.58 | 3343.05 | 382.35 | 21.4% (15) |
| EVPM | 74,301.96 | 1018.95 | 772.52 | 4359.14 | 421.93 | 40.0% (28) |
| EVM | 162,308.58 | 1355.02 | 371.92 | 92,365.76 | 1319.67 | 0.0% (0) |

*EVPRM* expected value with proxy and repetition bias model, *EVPBM* Expected value with proxy and default bias model, *EVPM* expected value with proxy model, *EVM* expected value model, *LOOIC* leave-one-out information criterion (lower values indicate higher predictive accuracy), *SE* standard error of LOOIC, *pLOOIC* effective number of parameters penalty, *dLOOIC* LOOIC difference relative to the model with highest predictive accuracy, i.e., lowest LOOIC value, *dSE* standard error of dLOOIC based on point-wise estimates, *Best Fit* number of participants whose behavior was best explained by the model. *N* = 70 participants.

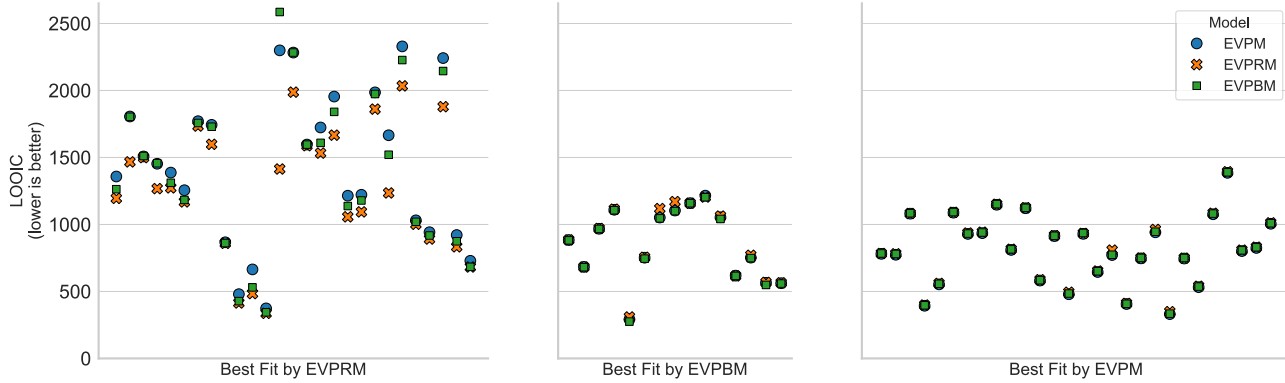

**Fig. 5 | Model comparison at participant level.** Predictive accuracy indicated by leave-one-out information criterion (LOOIC) for each participant and each model. The LOOIC values for expected value with proxy and repetition bias model (EVPRM), expected value with proxy and default bias model (EVPBM), and expected value with proxy model (EVPM) for each participant are aligned vertically. Participants are grouped depending on which model showed the highest predictive accuracy. LOOIC values for the expected value model (EVM) are not depicted, because of the low predictive accuracy of the EVM. EVPRM: *n* = 27 participants, EVPBM: *n* = 15 participants, EVPM: *n* = 28 participants.

behavior was best described by either EVPBM or EVPM did not show a clear difference in LOOICs. This indicates that these three models explained behavior equally well. As the participants fitted best by EVPBM and EVPM were found to have low repetition bias strength in the EVPRM (see next paragraph and Fig. 6), the EVPRM and the two alternative models are, for these participants, practically mathematically equivalent. We conclude that the EVPRM is the best model for 27 of the participants and is as good as the other two models for the remaining 43 participants.

Next we assessed if participants best fitted by EVPRM are the participants with a high repetition bias strength (see Supplementary Fig. 2A). To do this, we analyzed the distribution of the inferred parameter values of the EVPRM and grouped participants based on the model that explained their behavior best (see Fig. 6). The Shapiro-Wilk tests indicated a significant violation from normality, EVPRM: $W = 0.79$, $p < 0.001$, EVPBM: $W = 0.65$, $p < 0.001$, EVPM: $W = 0.38$, $p < 0.001$. Therefore, a Kruskal-Wallis *H*-test for independent samples was conducted, $H(2) = 47.66$, $p < 0.001$, $\eta^2 = 0.68$, indicating significant differences in repetition bias strengths between groups (see Fig. 6A). To further identify significant differences between groups, post-hoc pairwise comparisons using Conover's test were conducted. As expected, participants whose behavior was best explained by EVPRM showed significantly higher inferred repetition bias strengths than EVPBM ($p < 0.001$) and EVPM ($p < 0.001$).

Because we fitted all models with all sequences of actions, the repetition bias strength in EVPRM could have been dominated by repetition of the highlighted DAS. To exclude this possibility, we fitted an alternative EVPRM version with separate repetition bias strength parameters for the DAS and all other sequences. We found a strong significant correlation between the general repetition bias strength of the EVPRM and the repetition bias strength of the non-DAS sequences of the additional model, $r = 0.95$, $p < 0.001$, CI = [0.92, 0.97] (see Supplementary Fig. 1). We

concluded that the repetition bias strength *h* in the EVPRM was likely not primarily driven by the DAS but by other sequences.

Furthermore, we tested for group differences of inferred precision on expected rewards. The Shapiro-Wilk tests indicated a significant violation from normality for EVPRM, EVPRM: $W = 0.69$, $p < 0.001$, EVPBM: $W = 0.91$, $p = 0.123$, EVPM: $W = 0.96$, $p = 0.314$. Therefore, a Kruskal-Wallis *H*-test for independent samples was conducted, $H(2) = 31.61$, $p < 0.001$, $\eta^2 = 0.44$, indicating significant differences in precision on expected rewards between groups (see Fig. 6B). To further identify significant differences between groups, post-hoc pairwise comparisons using Conover's test were conducted. As expected, participants whose behavior was best explained by EVPM showed significantly higher inferred precision on expected rewards than participants best explained by EVPRM ($p < 0.001$).

Additionally, we tested for group differences of inferred approximated rewards. The Shapiro-Wilk tests indicated a significant violation from normality for EVPRM and EVM, EVPRM: $W = 0.89$, $p = 0.009$, EVPBM: $W = 0.96$, $p = 0.609$, EVPM: $W = 0.91$, $p = 0.023$. Therefore, a Kruskal-Wallis *H*-test for independent samples was conducted, $H(2) = 6.09$, $p = 0.048$, $\eta^2 = 0.06$, indicating significant differences in approximated rewards between groups (see Fig. 6C). To further identify significant differences between groups, post-hoc pairwise comparisons using Conover's test were conducted. Here, only EVPRM and EVPBM showed significant differences ($p = 0.014$).

In what follows, we compare fitted model parameters with behavioral measures of performance. Given that the EVPRM model has the best fit for 27 participants, and fits the remaining participants as well as the other models, we limit our analyses to fitted parameters of the EVPRM.

**Increase in DAS usage in participants fitted best with EVPRM.** In our analyses based on summary measures above, we did not find a significant

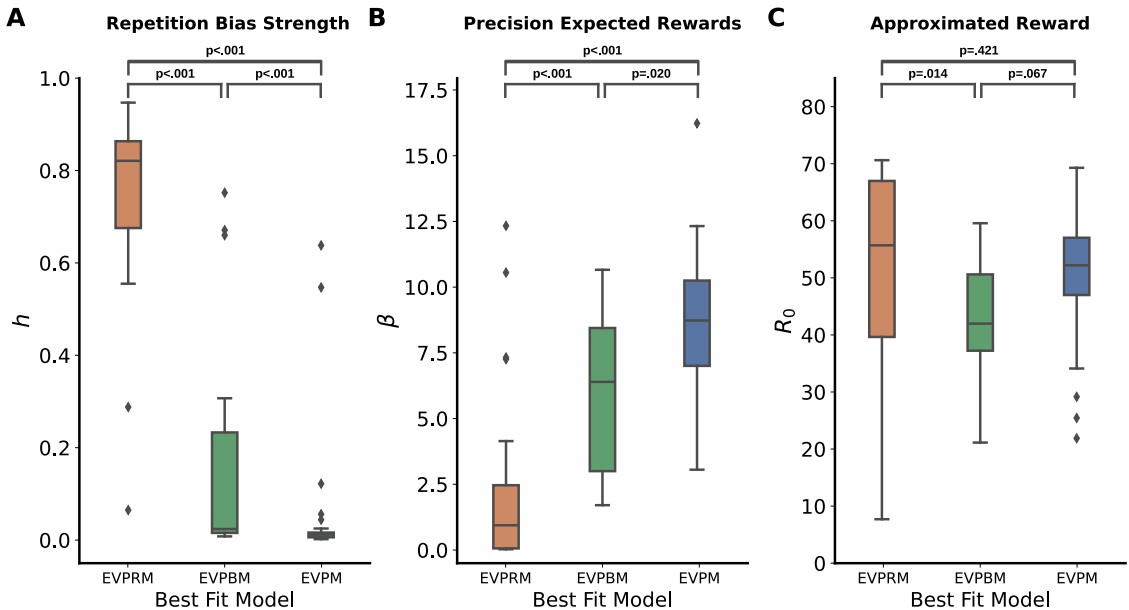

**Fig. 6 | Estimated parameter values of EVPRM partitioned by the model that explained participant behavior best.** Estimated value distribution for the three parameters of the EVPRM: **A** repetition bias strength $h$, **B** precision on expected rewards $\beta$, and **C** approximated reward $R_0$. Participants are partitioned according to the model with the lowest leave-one-out information criterion (LOOIC) value. Boxes represent the interquartile range (IQR). Horizontal lines inside boxes represent medians. Whiskers represent the 1.5 IQR of the lower and upper quartile. $p$ values represent Conover's post-hoc pairwise tests. EVPRM: $n = 27$ participants, EVPBM: $n = 15$ participants, EVPM: $n = 28$ participants.

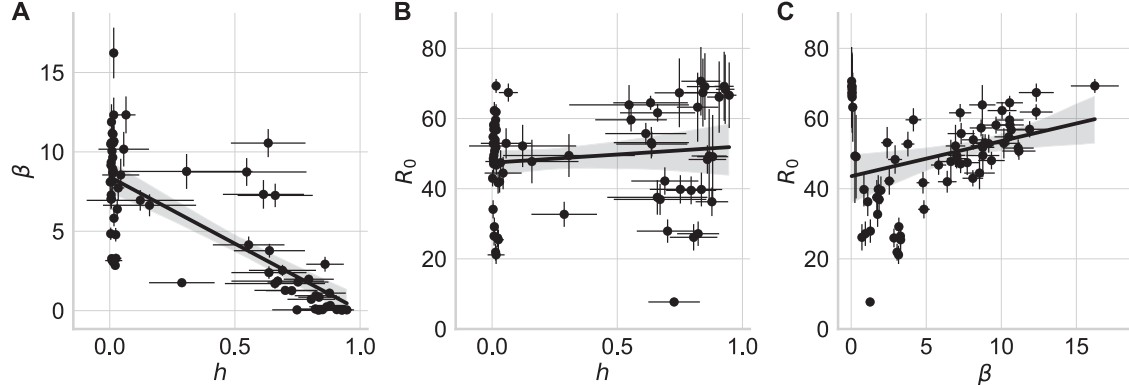

**Fig. 7 | Participant-level correlations between estimated parameters of the EVPRM. A** Correlation between repetition bias strength $h$ and precision over expected rewards $\beta$. **B** Correlation between repetition bias strength $h$ and approximated reward $R_0$. **C** Correlation between precision over expected rewards $\beta$ and approximated reward $R_0$. Thick solid lines represent linear regression model fitted to the data. Gray areas represent 95% confidence interval for the regression. Thin solid lines represent standard deviations (SD) of individual fitted parameter values. $N = 70$ participants.

increase of DAS usage from the first to the second half of the experiment over all participants. We repeated this analysis with only those 27 participants best fitted by the EVPRM. This group of participants showed a high repetition bias strength (see Fig. 6A), and as a consequence, we expected a significant increase of using the DAS from the first to the second half of the experiment for these participants. Indeed, the proportion of DAS choices of these participants significantly increased from the first half (46.2%, SD = 24.9%) to the second half (51.6%, SD = 26.1%) of the experiment, $t(26) = -2.27$, $p = 0.02$, $d = 0.21$, CI = $[-\infty, -0.01]$.

Furthermore, we tested if participants below the median of initial proportion of DAS choices have a greater repetition bias strength $h$ compared to participants with initial $p$(DAS) above the median. We could not find a significant difference using a two-sample $t$-test, $t(68) = -0.996$, $p = 0.322$, $d = 0.24$, CI = $[-0.27, 0.09]$. This means that we could not find evidence that the differences in repetition bias strength found by EVPRM are dependent of initial $p$(DAS).

**Correlations between parameters of EVPRM.** We analyzed the correlations between the parameter estimates for the three free model parameters repetition bias strength $h$, precision on expected rewards $\beta$, and approximated reward $R_0$. As these correlations are potentially not stable with our sample size[61], we use this analysis only as preliminary evidence for the expected tendencies to further validate our model.

As our model represents the influence of the repetition bias and expected rewards separately we can investigate the correlation between these two parameters. We expected that participants with a strong repetition bias strength $h$ are potentially more guided by past behavior than by expected rewards. Therefore, precision over expected rewards and repetition bias strength should show a negative correlation. We found such a significant negative correlation between the precision over expected rewards $\beta$ and the repetition bias strength $h$, $r = -0.75$, $p < 0.001$, CI = $[-0.84, -0.63]$ (see Fig. 7A). In addition, we found a significant positive correlation between $\beta$ and the approximated reward $R_0$, $r = 0.30$, $p = 0.012$, CI = $[0.07, 0.50]$ (see Fig. 7C).

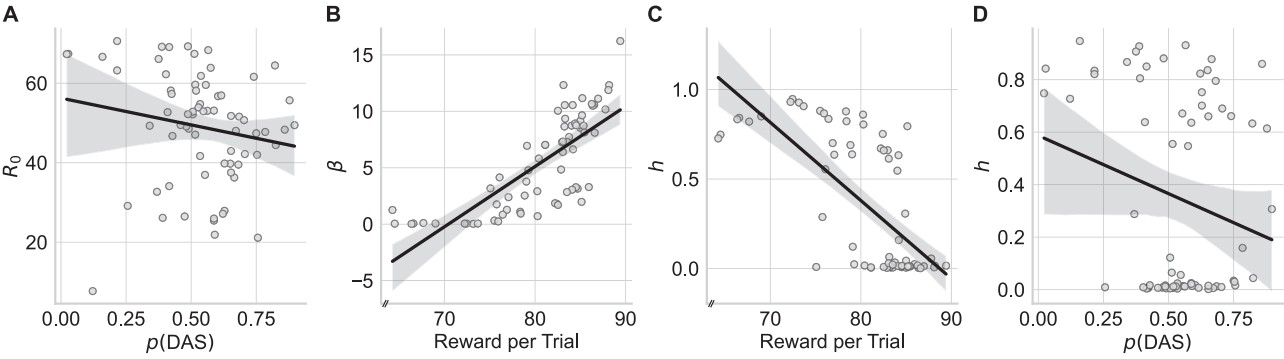

**Fig. 8 | Correlations between estimated parameters of expected value with proxy and repetition bias model (EVPRM) and performance measures. A** Correlation between free parameter of approximated reward $R_0$ and the mean proportion of DAS choices $p(DAS)$. **B** Correlation between model parameter of precision over expected rewards $\beta$ and mean reward per trial. **C** Correlation between model parameter repetition bias strength $h$ and mean reward per trial. **D** Correlation between model parameter repetition bias strength $h$ and mean reward per trial. Black solid lines represent linear regression model fitted to the data. Gray areas represent 95% confidence interval for the regression. $N = 70$ participants.

One reason for a strong repetition bias strength might be a low approximated reward. Therefore, the expectation of a low reward for unobserved sequences of actions could lead to stronger action repetition if participants can find an alternative sequence with higher reward. Contrary to our prediction, the approximated reward showed a positive correlation with the repetition bias strength, but this correlation was not significant, $r = 0.12$, $p = 0.303$, CI = [−0.11, 0.35] (see Fig. 7B).

**Correlations between model parameters of EVPRM with performance measures.** As a further intuitive validation measure, we also tested for correlations between the model parameters and performance measures. We expected that participants with a higher approximated reward $R_0$ would show a decreased reliance on the DAS, due to an expectation of higher rewards for alternative sequences of action. These participants should deviate from the DAS more frequently. Inferred values of $R_0$ correlated indeed negatively with the proportion of DAS choices $p(DAS)$, but this correlation was not significant, $r = -0.18$, $p = 0.142$, CI = [−0.40, 0.06] (see Fig. 8A).

Further, we expected that participants with higher precision over expected rewards $\beta$ were likely to earn more reward. This is because as $\beta$ increases, participants would have a lower uncertainty on the expected rewards. This increases the probability that participants select actions with higher expected rewards. As expected, participants showed a significant positive correlation between $\beta$ and the mean reward per trial, $r = 0.76$, $p < 0.001$, CI = [0.64, 0.85] (see Fig. 8B).

Crucially, we expected that participants with higher repetition bias strength $h$ would receive lower rewards because participants with a strong repetition bias tend to repeat past behavior rather than to maximize expected rewards. We found this significant negative correlation between the repetition bias strength and the mean reward obtained per trial, $r = -0.69$, $p < 0.001$, CI = [−0.80, −0.55] (see Fig. 8C). Accordingly, the achieved reward decreased with increasing repetition bias strength.

We also expected that participants with stronger repetition bias strength $h$ show shorter reaction times (RTs), as the repetition of past behavior should be executed faster than selecting yet unknown sequences of actions. Contrary to our expectation $h$ showed a significant positive correlation with mean RTs, $r = 0.37$, $p = 0.001$, CI = [0.15, 0.56]. We speculate that participants with a strong repetition bias were probably not as motivated as other participants and therefore slower in processing relevant stimuli and/or executing the movements. In combination with the tight deadline of six seconds, these participants probably relied more strongly on known sequences of actions. This speculation is supported by the significant positive correlation between the number of time out trials and the repetition bias strength.

Furthermore, although outside our prior hypotheses, we tested for a relationship between the repetition bias strength $h$ and the proportion of DAS choices $p(DAS)$. This correlation was negative but not significant, $r = -0.22$, $p = 0.069$, CI = [−0.43, 0.02] (see Fig. 8D). Therefore, we could not find evidence that the strength of the repetition bias had an influence on the general propensity to use the DAS. As the repetition bias parameter represents the increasing frequency-based influence of frequently repeated sequences of actions (not only the DAS), a correlation with the general proportion of DAS choices is not expected, but rather a relationship between the repetition bias parameter and the increase in DAS choices throughout the experiment. Therefore, frequent repetition of the DAS is indeed a prerequisite to develop a repetition bias for the DAS, but the general proportion of DAS choices throughout the experiment does not indicate a repetition-based increase of DAS choices over the course of the experiment. See Supplementary Table 2 for all correlational analyses.

## Discussion

In this study, we have shown experimental evidence of a repetition bias that increases the probability of performing a sequence of actions as a function of how frequent this action sequence had been used before, over the course of the experiment. To show this repetition bias, we introduced a sequential decision-making task and employed a recently proposed computational model that describes action selection as a balance between goal-directed action and a repetition bias.

We developed the Y-navigation (Y-NAT) task, where participants had to meet trial-specific goals by collecting points. We gave participants information about a default action sequence (DAS) that let them obtain the maximum expected reward in nearly half of all trials. With this manipulation we ensured that participants repeat at least one sequence of actions frequently.

In our behavioral analyses, we found that nearly all of our participants (94%) used the DAS most frequently, among all possible sequences. Participants executed the DAS even when the DAS did not provide the highest reward. However, our subsequent summary measures analyses to test for a repetition-induced increase in DAS usage only revealed non-significant trends, and the evidence remained inconclusive.

We complemented our analyses by a computational modeling approach to tease apart effects of reward seeking and repetition. Using four different models, we assessed whether explicit modeling of repetition learning over the course of the experiment reveals a repetition bias. Indeed, the repetition bias model, which we called the expected value with proxy and repetition bias model (EVPRM), explained participants' behavior best. Furthermore, we found that the repetition bias strength was negatively

related to task performance, suggesting an opposition between goal-directed performance and repetition bias strength.

With the proposed model-based approach we were able to rule out several alternative explanations for the observed effects. First, we can exclude random responding, as a random response model had a very low predictive accuracy, and all participants either relied on expected rewards or showed a repetition bias. Second, we excluded behavioral repetition as a fixed-choice strategy not influenced by past actions. For instance, one strategy could be to always select the incentivized DAS. As the DAS was of high value during the initial few trials of each block and always resulted in a reward, consistently repeating the DAS would classify as goal-directed behavior and would not be evidence for a repetition bias. We tested this alternative using a model that replaced the repetition bias effect by a constant bias added to the expected reward for the DAS, leading to constant higher choice probabilities for the DAS over the course of the experiment. Using model comparison we ruled out this alternative. Further evidence against a constant but not increasing influence of repetition is the finding of a general reduction in behavioral variability over time. Third, we used an alternative implementation of the winning EVPRM model to show that the repetition bias strength is not primarily driven by the highlighted DAS but also by repetition of other sequences.

The finding that the behavior of only a subgroup (ca. 39%) of participants was best explained by the repetition bias model seems consistent with previous studies where only subsets of participants were found to show habitual behavior[62,63]. One reason might be a strong motivation to perform well in experimental tasks[64]. This motivation probably prompts participants to use goal-directed behavior to collect reward. This effect might be strengthened by our performance-based bonus payment, as incentives have been shown to modulate cognitive effort[65]. This interpretation is also consistent with the finding that, contrary to our prior expectation, participants with an increased repetition bias strength showed slower RTs. Slower RTs are related to poorer performance and could be an indicator of less motivation and thus less goal-directed behavior for participants with strong repetition bias.

### Repetition bias and cost-benefit arbitration

In our task, like in many others, the effect of a potential repetition bias can only be measured in combination with concurrent goal-directed behavior. Specifically, according to the model, the first few decisions in a new task context are mainly based on expected rewards. Concurrently, the effect of the repetition bias ramps up and has, as we found, a measurable effect on action selection. While this is the concrete mechanism in the present model (see also ref. 42) the increasing influence of the repetition bias could also be viewed as an efficient, dynamic cost-benefit arbitration. MB planning is associated with cognitive costs[66,67], and it has been postulated that decision makers compute whether it is worth investing the cognitive effort. It might be that the inferred repetition bias strength is just a measurable expression of such a cost-benefit arbitration.

An alternative view is to turn this argument around and to postulate that the computation and use of the repetition bias is the causal underlying mechanism, which is observed and eventually interpreted as an apparent dynamic cost-benefit analysis. What speaks for this view is that the repetition bias is simple to compute because the model just increases a task-specific counter by 1. In the brain, this would correspond to a simple strengthening of a context-action association. Conversely, it has been shown that, in principle, cost-benefit arbitration leads to a computationally involved recursive planning process[66]. The question, which can be addressed in future studies, is whether a simple repetition bias computation is enough to explain apparent cost-benefit computations to generate behavior.

### Relation to other models and implications for habit learning

Although the idea of a value-free repetition bias has been proposed in psychology at least more than a century ago[1], it does not seem to play much of a role in explaining or modeling value-based decisions currently, beyond the recent studies by ref. 41 and ref. 42. However, the repetition bias is related

to the well-established stimulus-response (S-R) learning, as repetition facilitates the formation of S-R associations and recency effects in value-based decision-making tasks[26,27,68]. In addition, a recent study in visual perception proposed a choice history model, which is congruent with a repetition bias[21]. Behavioral repetition of action sequences has also been identified as a way to optimize the trade-off between maximizing reward and a reduction of policy complexity[43].

Importantly, the repetition learning mechanism is different from stimulus-response associations typically found in devaluation studies (e.g., [46,69,70]), and different from a potential trade-off between MF and MB RL[29,71], because, in contrast to MF RL, the repetition bias is value-free[41] and does not directly depend on past rewards.

Frequent repetition is considered a crucial prerequisite for habit formation[26,27]. The repetition bias decreased the influence of goal-directed control with behavioral repetition. Therefore, the repetition bias effect could potentially contribute to the development of habits. This suggests that this mechanism and its predictions could be used to investigate habit learning in early phases and disambiguate from effects due to goal-directed control. Especially, concerning the lack of a unified methodology for measuring habits[27,59], our task and the repetition bias could in principle be used to measure the tendency towards habitual behavior during only a few hundred trials without the need to implement habitual learning with over thousands of trials[46,47,62,72] and sessions over 2[47] to 4[46] days.

Indeed, many studies investigated the influence of repetition through habits. In these studies habits are typically only measured indirectly, as a lack of goal-directed behavior during an extinction phase[27,73]. However, a lack of goal-directed behavior can alternatively emerge due to an inaccurate representation of action-outcome contingencies during extinction[74], or random responding due to a lack of motivation[27]. Instead, here we measured repetitive behavior directly through a combination of task design and a model-based approach, enabling us to measure positive characteristics of repetitive behavior. Additionally, our task did not consist of separate training and extinction phases, and we provided feedback after each trial. This approach avoids a potentially inaccurate representation of the expected rewards.

### Limitations

A potential limitation of our model is the limited identifiability of the repetition bias strength with overestimation of low to medium strengths ($h < 0.40$) in some cases and underestimation of high strengths ($h > 0.80$) (see Supplementary Fig. 4A). However, most of the participants showed very low or medium to high repetition bias strengths, where a strong underestimation of low values or a strong overestimation of high values is unlikely. A stronger independence of repetition and expected rewards within the task could improve the accuracy of inference of the repetition bias strength.

Consistent with model assumptions, participants best fitted by EVPRM showed stronger repetition biases and an increase of DAS choice proportions from the first to the second half of the experiment. Note that this group also had a below-average proportion of DAS choices during the first half of the experiment (46.2% vs 53.3%). One potential explanation for this finding is that these participants with high repetition bias strength were in general less goal-directed and made more exploratory choices during the initial trials of the task. Subsequently, based on their strong repetition bias and after learning that there is a high proportion of trials where the DAS was one of the sequences with the highest expected reward, they increased their DAS choice proportions more steeply than other participants. However, the design of our task does not allow us to directly test this hypothesis.

### Conclusion

In conclusion, we introduced a sequential decision making task, with which we demonstrated the influence of both expected rewards and a repetition bias on decision making. Using computational modeling we provided empirical evidence for a repetition bias which is simply expressed as a value-free increase of choice probability each time an action sequence is

performed. This repetition bias mechanism emphasizes the importance of considering frequency-based mechanisms besides reward-driven mechanisms in future studies.

## Data availability
The raw datasets and processed dataset for this study are available at OSF: https://doi.org/10.17605/OSF.IO/RH42A[75].

## Code availability
The code for running the experiment and the Python code used for analysis are available at OSF: https://doi.org/10.17605/OSF.IO/RH42A[75].

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

## Acknowledgements

This work was supported by the German Research Foundation (DFG, Deutsche Forschungsgemeinschaft), SFB 940-Project number 178833530 to B.J.W. and S.K. and TRR 265-Project number 402170461 to S.S. and S.K. and as part of Germany's Excellence Strategy-EXC 2050/1-Project number 390696704-Cluster of Excellence, Centre for Tactile Internet with Human-in-the-Loop (CeTI) of TUD Dresden University of Technology to D.C.R. and S.K. The funders had no role in study design, data collection and analysis, decision to publish or preparation of the manuscript.

## Author contributions

Eric Legler: Conceptualization, Methodology, Data collection, Formal analysis, Visualization, Writing—original draft, Writing—review and editing. Darío Cuevas Rivera: Methodology, Formal analysis, Writing—original draft, Writing—review and editing. Sarah Schwöbel: Methodology, Writing–review and editing. Ben J. Wagner: Writing—original draft, Writing–review and editing. Stefan Kiebel: Conceptualization, Writing—original draft, Writing—review and editing, Supervision, Funding acquisition.

## Funding

## Competing interests

The authors declare no competing interests.
