## [Transparent Peer Review file · Communications Psychology]

Cognitive Computational Model Reveals Repetition Bias in a Sequential Decision-Making Task

Corresponding Author: Mr Eric Legler

Version 0:

Decision Letter:

Dear Mr Legler,

Thank you for your patience during the peer-review process. Your manuscript titled "Cognitive Computational Model Reveals Repetition Bias in a Sequential Decision-Making Task" has now been seen by 2 reviewers, and I include their comments at the end of this message. They find your work of interest but raised some important points. We are interested in the possibility of publishing your study in Communications Psychology, but would like to consider your responses to these concerns and assess a revised manuscript before we make a final decision on publication.

We therefore invite you to revise and resubmit your manuscript, along with a point-by-point response to the reviewers. Please highlight all changes in the manuscript text file.

Editorially, we consider it critical that the revised manuscript addresses reviewer 1's methodological concerns, including those regarding the relationship between the model parameters and the proportion of default action sequence choices. In this regard, please note that we do not permit interpretation of the absence of a statistically significant result as evidence for the absence of an effect. Strong claims about the absence of an effect or difference require Bayesian statistics or equivalence tests. Please provide a sensitivity analysis (not a post-hoc power analysis) to report the sensitivity of the design to the smallest effect size of interest (not the established effect size). Reviewer #2 also requests a more comprehensive report of participant demographics, which aligns with our guidelines (see Reporting Summary). Finally, please ensure that the presentation is clear and comprehensive throughout- in particular, there is no limit on the length of the Methods section and all key results should be in the main manuscript.

I am attaching an Editorial Requests Table that details critical reporting requirements for the revised manuscript. Please attend to each item and ensure your manuscript is fully compliant. We are requesting that your manuscript aligns with these requirements as this facilitates the evaluation of your manuscript, reducing delays in re-review and potential future acceptance. If your revised manuscript is not aligned with these requests on major issues, such as those concerning statistics, it may be returned to you for further revisions without re-review. Additional information can be found in our style and formatting guide Communications Psychology formatting guide.

Please use the following link to submit your

- revised manuscript,
- point-by-point response to the referees' comments,
- cover letter (as a separate document),
- the Editorial Policy Checklist (see below),
- the Reporting Summary (see below), and
- the completed Editorial Request Table (attached):

Link Redacted

Best regards,

Troby Lui

Troby Lui, PhD
Associate Editor
Communications Psychology

REVIEWER EXPERTISE:

Reviewer #1: reward learning, value-based decision-making, reinforcement learning

Reviewer #2: habit, value-based decision-making, reinforcement learning

REVIEWER REPORTS:

Reviewer #1 (Remarks to the Author):

This paper reports data on a novel sequential decision-making task, which, coupled with a computational model, reveals a repetition bias tendency that could interfere with goal-directed (value-based) decision-making. The key results of this study include the identification of a repetition bias parameter that is sensitive to the tendency to repeat action sequences, which is thought to capture a habit-related construct (i.e., a value-free process).

I think this paper is interesting and potentially makes a theoretical contribution to the domain of goal-directed behaviour, namely, to provide empirical data for the influence of repetition in action selection. There are several noteworthy strengths: the development of the task was sufficiently motivated by past literature; I thought the authors have conducted a comprehensive set of analyses on model agnostic and model-based variables; parameter recovery and posterior predictive simulations to further validate their model are a strength; their reporting was also sufficiently clear and transparent for replication; the authors have also made their data and code publicly available. However, I have some concerns about the evidence for the primary claims made in this manuscript, the complexity of the task, the construct validity, and the inferences on habitual mechanisms. I hope the comments would be useful to the authors.

1) The lack of relationship between a key computational parameter and model-agnostic measure: The central claim by the authors is that they 'shown experimental evidence of a repetition bias that increases the probability of performing a sequence of actions as a function of how frequent this action sequence had been used before' (lines 371 –373). Although several analyses seem to be consistent with this, one key omission is the examination of the relationship between this computational parameter and model-agnostic measures of repetition bias, i.e., the probability of the default action sequence, $p(\text{DAS})$. The authors relegated this information to the supplementary material, which I note that it is not statistically significant. On face value, the lack of relationship weakens the authors' central claims. Could the authors comment on the lack of this relationship in the manuscript, explaining whether this affects the interpretation of the computational parameter? It would be important to address this point as it relates to their key findings.

2) Parameter identifiability: The precision over expected reward, β , and the repetition bias, h , parameters are quite strongly correlated with one another. Whilst the authors report this as consistent with theoretical predictions, I am worried about the identifiability of the parameter during the model fitting procedure, i.e., whether one parameter is trading off the other. Indeed, the parameter recovery plots (supplementary figure 2) revealed that the recovery for β looks much better than for h . To enhance the reliability of this parameter, the authors would need to clarify whether the parameters are indeed identifiable, and/or whether this is a problem with the model fitting procedure / model itself.

3) Task complexity: From my reading, the sequential decision-making task seems to be a very complex task. I had to read the methods several times to somewhat grasp the design of the task. Do the authors have any tests / measures / ratings to confirm that participants indeed understood the instructions and were engaging in the task meaningfully? Reporting these would be crucial to establish confidence in the task. If not, the authors might need to acknowledge this in the limitations section.

4) The question of construct validity: the authors claimed that the repetition bias identified through their task and computational model may underlie habit formation and made some inferences about 'value-free' habits. However, this assumes that the tendency to repeat action sequences causes one to be more habitual, which is not necessarily true. Did the authors collect any (direct or indirect) measures of habits, either behavioural or self-report, to support their claims? If not, then the authors would need to rephrase their claims, and acknowledge this as an important limitation of their study.

5) Related point to 4), if no external habit measures were collected to verify construct validity, then the authors should rephrase or rewrite instances in their discussion where they use their data to infer about the mechanisms of habits, as their task alone neither tests for habit learning nor measures 'value-free' habits. These arguments need to be more nuanced and tentative.

6) Random response model: The authors claimed that random responding is unlikely (line 392), and I agree. To make this point more convincingly, I'd suggest the authors to include a computational model that simulates random responses and compare the fit of this model with the existing ones.

7) Participants were explicitly told about probabilistic bonus rewards for using the DAS in half of the blocks, which presumably would affect their behavioural tendency (e.g., repetition of DAS due to anticipated rewards instead of repetition bias). Could the authors clarify whether this design (i.e., the anticipation of unexpected rewards) was considered in the analyses?

8) Some comments on the statistical reporting and analyses:

a. In the 'Increase in DAS' section (page 7, lines 190 -- 196), I note that there were three trial types (DAS, partial DAS, non-DAS), but the authors have combined DAS and partial-DAS and conducted a one-sided t-test. I believe a one-way ANOVA would be more appropriate in this context.

b. Figure 2C: the authors would need to plot the results of the DAS-independent trials as well, to be consistent with the analyses.

c. Could the authors add the correlation coefficients to the parameter recovery (supplementary figure 2), either in the legends or the figure?

d. Could the authors add statistics to support their analyses in figure 5, either in the main text, legend, or the figure itself?

I also have some comments on the citations and writing, the latter being unclear at times:

1) The notion that repetition (i.e., overtraining) is a key element of habit formation originates from the seminal work by Dickinson and Adams; these authors should be cited when this point is mentioned (e.g., line 53, line 70).

2) Related to 1), it would also be appropriate for the authors to allude to lack of human evidence for the effects of overtraining in increasing habit strength (i.e., Pool et al 2022) – a paper that the authors also cite in their manuscript.

3) Table 2: perhaps substitute 'best fit' with '% subjects with best fit' to make it clearer to the reader?

4) Line 464: the authors could consider citing Buabang et al 2023 (doi: 10.1037/xge0001280) for empirical data supporting the point on inaccurate representation of action-outcome contingencies during extinction.

5) I think there is a typo in the legends of Figure 11C: PMC instead of PMS.

6) I didn't understand 'However, the repetition bias could also decrease the probability to switch back to the DAS when the DAS is optimal later during the block.' Perhaps some rephrasing is needed?

7) Similarly, I also did not understand the sentence on lines 211 – 213: 'we calculated the correlation between the difference of used sequences and the difference of mean reward per trial between the first and the second half.'

Reviewer #2 (Remarks to the Author):

The authors report a study of repetition bias using a newly developed sequential decision-making task combined with computational modeling of choice behavior. They find evidence for a repetition bias in participants' behavior in addition to reward-oriented, goal-directed choice strategies, but also report inter-individual differences in how strongly participants rely on past choice frequency to inform their decisions. They discuss their findings in the context of habit learning. The manuscript is timely and relevant, well written, clear, and structured. The research question and hypotheses are well informed by the literature and the methods are sound and well-chosen to answer the question at hand. The description of the computational models used are clear and comprehensible. I want to commend the authors for openly sharing their data, materials, and analysis code. I have read the paper with enthusiasm and offer only a few minor points I would like to invite the authors to address.

1. One problem of previous experimental approaches to the induction and measurement of frequency-based behavioral effects is that often the magnitude of reinforcement and choice frequency are correlated by design. Thus, participants have an incentive to choose options more often leading to more reward. You seem to have mitigated this effect by your task design giving participants instead an incentive to reach a specified, changing level of reward, yet I would like you to discuss this point more explicitly in your manuscript drawing more attention to this issue.

2. The Participants section of your Methods does not state how you decided on the aimed-at sample size. Did you perform a sample size analysis prior to data assessment? If so, please state this explicitly in the Methods. If not, please provide your thoughts in the process of determining the sample size.

3. The sample is imbalanced regarding gender (50 female out of 70 participants). I would not expect sex to have an

influence on this basic learning process, yet a discussion of this point (and other sample size characteristics like the ethnicity, socioeconomic status, and education level of participants) would be valuable for the later integration and generalization of your findings.

4. The correlational analyses you present in Figures 3C and 6 are probably underpowered (Kretzschmar & Gignac, 2019; Schönbrodt & Perugini, 2013) and, at least in the case of the results in Fig. 3C, seem influenced by a few influential data points in the upper left quadrant. Please discuss this shortcoming in terms of statistical power and use caution when interpreting the correlational effects.

5. The 27 participants best described by the EVPRM seem to start with an especially low proportion of DAS choices (46.2% choice of the DAS in the first half to 51.6% in the second half (ll. 324-326) vs. 53.3% in the first half to 55.2% in the second half for the whole sample; ll. 146-148). In how far might the results of your model-free and model-based analyses be influenced by differences in starting point rather than an increased tendency to repeat DAS? Might this point towards difference in how participants approach the task or understand the instructions even before behavioral frequency can have an effect on behavior?

References

- Kretzschmar, A., & Gignac, G. E. (2019). At what sample size do latent variable correlations stabilize? *Journal of Research in Personality*, 80, 17–22. <https://doi.org/10.1016/j.jrp.2019.03.007>
- Schönbrodt, F. D., & Perugini, M. (2013). At what sample size do correlations stabilize? *Journal of Research in Personality*, 47(5), 609–612. <https://doi.org/10.1016/j.jrp.2013.05.009>

EDITORIAL POLICIES

We ask that you ensure your manuscript complies with our editorial policies and reporting requirements.

To that end, we require revised manuscripts to be accompanied by two completed items: a reporting summary that collects information on study design and procedure, and an editorial policy checklist that verifies compliance with all required editorial policies.

- <https://www.nature.com/documents/nr-reporting-summary.zip>>Nature Research Reporting Summary
- <https://www.nature.com/documents/nr-editorial-policy-checklist.pdf>>Editorial Policy Checklist

All points on the policy checklist must be addressed. Your revised manuscript can only be sent back to the referees if these checklists are completed and uploaded with the revision.

Notes: If you have submitted a Stage 1 Registered Report, Review, Primer, Comment, or Perspective you do not need to submit these forms. If you have already submitted these forms, you may disregard this request.

Version 1:

Decision Letter:

Dear Mr Legler,

Your manuscript titled "Cognitive Computational Model Reveals Repetition Bias in a Sequential Decision-Making Task" has now been seen by our reviewers, whose comments appear below. In light of their advice I am delighted to say that we are happy, in principle, to publish a suitably revised version in Communications Psychology.

We therefore invite you to revise your paper one last time to address the remaining concerns of our reviewers and a list of editorial requests. At the same time we ask that you edit your manuscript to comply with our format requirements and to maximise the accessibility and therefore the impact of your work.

EDITORIAL REQUESTS:

SUBMISSION INFORMATION:

OPEN ACCESS:

*** TRANSPARENT PEER REVIEW:** Communications Psychology uses a transparent peer review system. On author request, confidential information and data can be removed from the published reviewer reports and rebuttal letters prior to publication. If you are concerned about the release of confidential data, please let us know specifically what information you would like to have removed. Please note that we cannot incorporate redactions for any other reasons.

*** CODE AVAILABILITY:** All Communications Psychology manuscripts must include a section titled "Code Availability" at the end of the methods section. We require that the custom analysis code supporting your conclusions is made available in a publicly accessible repository at this stage; please choose a repository that generates a digital object identifier (DOI) for the code; the link to the repository and the DOI must be included in the Code Availability statement. Publication as Supplementary Information will not suffice.

*** DATA AVAILABILITY:**

Link Redacted

Best regards,

Troby Lui

Troby Lui, PhD
Associate Editor
Communications Psychology

REVIEWERS' COMMENTS:

Reviewer #1 (Remarks to the Author):

Thank you for the revised manuscript. I commend the authors for thoroughly and carefully carrying out the revisions, which, in my view, have strengthened the manuscript considerably. I have no further comments and am pleased to support the publication, as it makes a unique contribution to the wider literature on goal-directed decision-making.

Reviewer #2 (Remarks to the Author):

The authors have addressed all my comments very well and I currently see no further issues with this submission. Thank you for being so responsive.

Reviewer #1:

We thank the reviewer for his/her valuable and constructive advice when reviewing our manuscript. We addressed all the comments.

Note, that line numbers refer to the new manuscript file **without** marked changes. We added two versions, one with, and the other without marked changes.

Reviewer #1, initial comment:

This paper reports data on a novel sequential decision-making task, which, coupled with a computational model, reveals a repetition bias tendency that could interfere with goal-directed (value-based) decision-making. The key results of this study include the identification of a repetition bias parameter that is sensitive to the tendency to repeat action sequences, which is thought to capture a habit-related construct (i.e., a value-free process).

I think this paper is interesting and potentially makes a theoretical contribution to the domain of goal-directed behaviour, namely, to provide empirical data for the influence of repetition in action selection. There are several noteworthy strengths: the development of the task was sufficiently motivated by past literature; I thought the authors have conducted a comprehensive set of analyses on model agnostic and model-based variables; parameter recovery and posterior predictive simulations to further validate their model are a strength; their reporting was also sufficiently clear and transparent for replication; the authors have also made their data and code publicly available. However, I have some concerns about the evidence for the primary claims made in this manuscript, the complexity of the task, the construct validity, and the inferences on habitual mechanisms. I hope the comments would be useful to the authors.

Reviewer #1, comment 1:

1) The lack of relationship between a key computational parameter and model-agnostic measure: The central claim by the authors is that they 'shown experimental evidence of a repetition bias that increases the probability of performing a sequence of actions as a function of how frequent this action sequence had been used before' (lines 371 –373). Although several analyses seem to be consistent with this, one key omission is the examination of the relationship between this computational parameter and model-agnostic measures of repetition bias, i.e., the probability of the default action sequence, $p(\text{DAS})$. The authors relegated this information to the supplementary material, which I note that it is not statistically significant. On face value, the lack of relationship weakens the authors' central claims. Could the authors comment on the lack of this relationship in the manuscript, explaining whether this affects the interpretation of the computational parameter? It would be important to address this point as it relates to their key findings.

Response to Reviewer #1, comment 1:

Yes, we agree that the discussion of the non-significant correlation between the repetition bias parameter h and the proportion of DAS choices $p(\text{DAS})$ should be included in the main text for completeness. However, note that this relationship was not one of our prior hypotheses. The reason is that we rather hypothesized, based on our central claim, a significant correlation between the increase in $p(\text{DAS})$, over the course of the experiment, with repetition bias strength h . This fits with the cited central claim, where we postulate a

mechanism which increases the probability to select the DAS per repetition. The correlation between the mean $p(\text{DAS})$ and parameter h would not be necessarily expected under our central claim. Likewise, a lack of such a significant relationship does not affect our central claim. Following the reviewer's suggestion, we have added the correlation to Figure 11, and added the paragraph below to the section *Correlations between model parameters of EVPRM with performance measures* of the manuscript (manuscript file without marked changes, line 752).

“Furthermore, although outside our prior hypotheses, we tested for a relationship between the repetition bias parameter h and the proportion of DAS choices $p(\text{DAS})$. This correlation was negative but not significant, $r = -.22, p = .069, CI = [-.43, .02]$ (see Figure 11D). Therefore, we could not find evidence that the strength of the repetition bias had an influence on the general propensity to use the DAS. As the repetition bias parameter represents the increasing frequency-based influence of frequently repeated sequences of actions (not only the DAS), a correlation with the general proportion of DAS choices is not expected, but rather a relationship between the repetition bias parameter and the increase in DAS choices throughout the experiment. Therefore, frequent repetition of the DAS is indeed a prerequisite to develop a repetition bias for the DAS, but the general proportion of DAS choices throughout the experiment does not indicate a repetition-based increase of DAS choices over the course of the experiment.”

Reviewer #1, comment 2:

2) *Parameter identifiability: The precision over expected reward, β , and the repetition bias, h , parameters are quite strongly correlated with one another. Whilst the authors report this as consistent with theoretical predictions, I am worried about the identifiability of the parameter during the model fitting procedure, i.e., whether one parameter is trading off the other. Indeed, the parameter recovery plots (supplementary figure 2) revealed that the recovery for β looks much better than for h . To enhance the reliability of this parameter, the authors would need to clarify whether the parameters are indeed identifiable, and/or whether this is a problem with the model fitting procedure / model itself.*

Response to Reviewer #1, comment 2:

We agree with the reviewer that reliability of the repetition bias parameter h would in general be a problem of interpretation of the inferred parameter h . However, we found in practice inferred repetition bias strengths h mainly at the extremes. With the parameter recovery we could not find hints that medium or high repetition bias strengths were strongly underestimated, increasing the reliability of the inferred low h values. Similarly, low

repetition bias strengths were partially overestimated, but not above .65, and most participants with strong repetition bias have an inferred h value above .65. This strengthens the reliability of inferred strong repetition bias. We added this limitation to the *Limitations* section (line 870):

“A potential limitation of our model is the limited identifiability of the repetition bias parameter with overestimation of low to medium strengths ($h < .40$) in some cases and underestimation of high strengths ($h > .80$) (see Supplementary Figure 2A). However, most of the participants showed very low or medium to high repetition bias strengths, where a strong underestimation of low values or a strong overestimation of high values is unlikely.”

Reviewer #1, comment 3:

3) Task complexity: From my reading, the sequential decision-making task seems to be a very complex task. I had to read the methods several times to somewhat grasp the design of the task. Do the authors have any tests / measures / ratings to confirm that participants indeed understood the instructions and were engaging in the task meaningfully? Reporting these would be crucial to establish confidence in the task. If not, the authors might need to acknowledge this in the limitations section.

Response to Reviewer #1, comment 3:

Yes, we concur that the task description is more on the complex side. Therefore, during task development and piloting we established five measures to ensure that participants understand the task and that we only analyzed data for which there was evidence that participants understood the task. First, to motivate participants to learn this complex task, we made sure that participants perceived it as engaging by a game-like structure and goal. Second, we used an elaborate instruction phase, where all relevant features of the task were introduced in a step-by-step manner (see *Experimental Task*). Third, to identify participants who did not understand the task, following the completion of the main task, participants were asked about problems when performing the task and the strategy they employed. We could not find any evidence in these responses that participants did not understand the task instructions. We added participants' answers to the GitHub repository. Fourth, based on the comparison with a random model agent, we concluded that participants' behavior is goal-directed and based on a sufficient understanding of the task. Fifth, we excluded 4 participants with $p(\text{DAS}) > .90$. Although this behavior is per se not incongruent with task instructions, we assumed that these participants, for some reason, did not fully understand or did not want to follow the goal-directed nature of the task. In conclusion, we believe that the remaining participants understood the task.

We added the file containing the responses of all participants to the repository, added a paragraph about the responses, and integrated the exclusion criteria into this new paragraph at the *Experimental Task* section (line 225):

“To ensure that participants understood the instructions of the task, we immediately asked them to describe their strategies and report any issues upon task completion. The majority of participants reported either no difficulties or only minor technical problems during a few trials. No participant indicated that they did not understand

the task (all responses are available on GitHub). When asked about the strategy they used, most participants described meaningful strategies. For instance, 22 participants reported a strategy of using the DAS within a specified range of point difference between the goal and expected points obtained by using the DAS, and searching for alternative sequences outside this range. Additionally, we excluded participants with a lack of behavioral variability (i.e., proportion of DAS choices $>.90$), as we assumed that these participants did not fully understand the goal-directed nature of the task. All remaining participants achieved a higher average reward per trial (all >64) than what would be achieved by responding randomly (50.43).”

Reviewer #1, comment 4:

4) The question of construct validity: the authors claimed that the repetition bias identified through their task and computational model may underlie habit formation and made some inferences about ‘value-free’ habits. However, this assumes that the tendency to repeat action sequences causes one to be more habitual, which is not necessarily true. Did the authors collect any (direct or indirect) measures of habits, either behavioural or self-report, to support their claims? If not, then the authors would need to rephrase their claims, and acknowledge this as an important limitation of their study.

Response to Reviewer #1, comment 4:

Indeed, we did not collect any measures of habit and rephrased the text following the suggestion of the reviewer. Clearly, we did not aim at showing that the repetition bias mechanism may be the basis for habit learning but used this speculation as motivation in the *Introduction* and later came back to this point in the *Discussion* (see also 5). We agree with the reviewer that rephrasing the text in the *Introduction* is necessary to clarify that this point was not of our claims. We have changed the last paragraph of the *Introduction* (line 77) to emphasize and clarify that our sole focus was on repetition, and not on a relation to habit formation.

“Here, we followed this theoretical lead and assessed empirically, in human participants ($n = 70$), the effect of such a repetition bias on behavior. Our goal was to investigate the mutual influence of rewards and repetition on task behavior. Note that our goal here was not to test hypotheses about habit formation.”

We also changed the *Conclusion* paragraph of the *Discussion* section, and do not mention habit learning any more (line 891):

“This repetition bias mechanism emphasizes the importance of considering frequency-based mechanisms besides reward-driven mechanisms in future studies.”

Reviewer #1, comment 5:

5) Related point to 4), if no external habit measures were collected to verify construct validity, then the authors should rephrase or rewrite instances in their discussion where they use their data to infer about the mechanisms of habits, as their task alone neither tests for habit learning nor measures ‘value-free’ habits. These arguments need to be more nuanced and tentative.

Response to Reviewer #1, comment 5:

Following the reviewer's suggestion, and related to our answer to point 4, we have rephrased the *Relation to other models and implications for habit learning* paragraph (line 848) of the *Discussion*.

"Frequent repetition is considered a crucial prerequisite for habit formation (Wood & Runger, 2016; Watson et al., 2022). The repetition bias decreased the influence of goal-directed control with behavioral repetition. Therefore, the repetition bias effect could potentially contribute to the development of habit formation. This suggests that this mechanism and its predictions could be used to investigate habit learning in early phases and disambiguate from effects due to goal-directed control."

Reviewer #1, comment 6:

6) *Random response model: The authors claimed that random responding is unlikely (line 392), and I agree. To make this point more convincingly, I'd suggest the authors to include a computational model that simulates random responses and compare the fit of this model with the existing ones.*

Response to Reviewer #1, comment 6:

We thank the reviewer for this suggestion. We implemented this random response model. As expected, the predictive accuracy of this model was significantly worse compared to the other models tested. We added these results to the section *Model Comparison* (line 633) and the *Discussion* (line 787):

"Because of its low predictive accuracy, we excluded the EVM from further analyses. Also, a random response model failed in explaining the data ($LOOIC = 158162.19, SE = 0.00$) and was not further considered."

"First, we can exclude random responding, as a random response model had a very low predictive accuracy, and all participants either relied on expected rewards or showed a repetition bias."

Reviewer #1, comment 7:

7) *Participants were explicitly told about probabilistic bonus rewards for using the DAS in half of the blocks, which presumably would affect their behavioural tendency (e.g., repetition of DAS due to anticipated rewards instead of repetition bias). Could the authors clarify whether this design (i.e., the anticipation of unexpected rewards) was considered in the analyses?*

Response to Reviewer #1, comment 7:

We agree with the reviewer that the bonus condition probably had an effect on participants' behavior. Therefore, we had calculated the standard descriptive statistics depending on the bonus condition but placed it into the supplementary material (Supplementary Table 1 in the original version). This analysis confirmed that in the bonus condition, $p(\text{DAS})$ and rewards were higher relative to the non-bonus condition. This bonus design is explicitly considered in all models analyses, including the EVPRM. We added the following sentence to the *Expected*

value with proxy and repetition bias model (EVPRM) section (line 273), to clarify and emphasize this.

“In addition, the expected reward of the DAS included the probabilistic bonus reward. Therefore, in all bonus trials the expected reward of the DAS was enhanced by 5 (probabilistic bonus of 20 with a probability of .25).”

Reviewer #1, comment 8a:

8) Some comments on the statistical reporting and analyses:

a. In the ‘Increase in DAS’ section (page 7, lines 190 -- 196), I note that there were three trial types (DAS, partial DAS, non-DAS), but the authors have combined DAS and partial-DAS and conducted a one-sided *t*-test. I believe a one-way ANOVA would be more appropriate in this context.

Response to Reviewer #1, comment 8a:

On reviewing the lines mentioned by the reviewer, we noticed that the phrasing was misleading. The description in the manuscript implied that we were comparing the proportion of partial DAS choices with that of non-DAS choices. What we did instead was to test for evidence for a weaker interpretation of the repetition bias. To do this, we combined DAS and partial DAS trials, assuming that DAS trials can be interpreted as special cases of partial DAS trials. We then conducted a *t*-test to test for a significant increase of the partial DAS (including DAS) trials from first to second half of the experiment. To clarify this, we have rephrased the second and third paragraphs of *Increase in DAS parts* in the manuscript (line 537) and updated the title of Figure 6C:

“To test for an increase in repeating the first move(s) of the DAS, we tested for differences in the proportion of partial DAS trials (including DAS trials) over halves and segments. A one-sided *t*-test for related samples revealed that the proportion of partial DAS (including DAS) trials significantly increased from the first half of the experiment to the second half, $t(69) = -6.46, p < .001, d = .49, CI = [-\infty, -0.06]$ (see Figure 6C). Similarly, a repeated measures ANOVA over segments showed a significant increase of partial DAS (including DAS) use over time, $F(3,207) = 19.53, p < .001, \eta_G^2 = .006$ (see Figure 6C).”

Reviewer #1, comment 8b:

b. Figure 2C: the authors would need to plot the results of the DAS-independent trials as well, to be consistent with the analyses.

Response to Reviewer #1, comment 8b:

Yes, we accordingly updated Figure 6C (see also 8a).

Reviewer #1, comment 8c:

c. Could the authors add the correlation coefficients to the parameter recovery (supplementary figure 2), either in the legends or the figure?

Response to Reviewer #1, comment 8c:

We followed the reviewer's suggestion and added the correlation coefficients to the figure.

Reviewer #1, comment 8d:

d. Could the authors add statistics to support their analyses in figure 5, either in the main text, legend, or the figure itself?

Response to Reviewer #1, comment 8d:

We added the corresponding statistics to the main text (lines 656, 674, and 682).

“Next we assessed if participants best fitted by EVPRM are the participants with a high repetition bias strength (see Figure 4A). To do this, we analyzed the distribution of the inferred parameter values of the EVPRM and grouped participants based on the model that explained their behavior best (see Figure 9). The Shapiro-Wilk tests indicated a significant violation from normality, EVPRM: $W = .79, p < .001$, EVPBM: $W = .65, p < .001$, EVPBM: $W = .38, p < .001$. Therefore, a Kruskal-Wallis H-test for independent samples was conducted, $H(2) = 47.66, p < .001, \eta^2 = 0.68$, indicating significant differences in repetition bias strengths between groups. To further identify significant differences between groups, post-hoc pairwise comparisons using Conover's test were conducted. As expected, participants whose behavior was best explained by EVPRM showed significantly higher inferred repetition bias strengths than EVPBM ($p < .001$) and EVPBM ($p < .001$).”

“Furthermore, we tested for group differences of inferred precision on expected rewards. The Shapiro-Wilk tests indicated a significant violation from normality for EVPRM, EVPRM: $W = .69, p < .001$, EVPBM: $W = .91, p = .123$, EVPBM: $W = .96, p = .314$. Therefore, a Kruskal-Wallis H-test for independent samples was conducted, $H(2) = 31.61, p < .001, \eta^2 = 0.44$, indicating significant differences in precision on expected rewards between groups. To further identify significant differences between groups, post-hoc pairwise comparisons using Conover's test were conducted. As expected, participants whose behavior was best explained by EVPBM showed significantly higher inferred precision on expected rewards than participants best explained by EVPRM ($p < .001$).”

“Additionally, we tested for group differences of inferred approximated rewards. The Shapiro-Wilk tests indicated a significant violation from normality for EVPRM and EVPBM, EVPRM: $W = .89, p = .009$, EVPBM: $W = .96, p = .609$, EVPBM: $W = .91, p = .023$. Therefore, a Kruskal-Wallis H-test for independent samples was conducted, $H(2) = 6.09, p = .048, \eta^2 = 0.06$, indicating significant differences in approximated rewards between groups. To further identify significant differences between groups, post-hoc pairwise comparisons using Conover's test were conducted. Here, only EVPRM and EVPBM showed significant differences ($p = .014$)”

Reviewer #1, comment 9.1:

I also have some comments on the citations and writing, the latter being unclear at times:

1) The notion that repetition (i.e., overtraining) is a key element of habit formation originates

from the seminal work by Dickinson and Adams; these authors should be cited when this point is mentioned (e.g., line 53, line 70).

Response to Reviewer #1, comment 9.1:

We agree and added the work by Dickinson & Adams as reference in the *Introduction* (line 52).

“Most importantly, repetitions are seen as a key element of habit formation (Adams and Dickinson, 1981; Adams, 1982; Wood and Runger, 2016; Watson et al., 2022).”

Reviewer #1, comment 9.2:

2) Related to 1), it would also be appropriate for the authors to allude to lack of human evidence for the effects of overtraining in increasing habit strength (i.e., Pool et al 2022) – a paper that the authors also cite in their manuscript.

Response to Reviewer #1, comment 9.2):

As the reviewer suggests, we added the paper from Pool et al. (2022) to the list of studies that investigate the effects of training duration on habit strength in the *Discussion* (line 853).

“Especially, concerning the lack of a unified methodology for measuring habits (Watson and De Wit, 2018; Watson et al., 2022), our task and the repetition bias could in principle be used to measure the tendency towards habitual behavior during only a few hundred trials without the need to implement habitual learning with over thousands of trials (Hardwick et al., 2019; Luque et al., 2020; Pool et al., 2022; Frolich et al., 2023) and sessions over two (Frolich et al., 2023) to four (Hardwick et al., 2019) days.”

Reviewer #1, comment 9.3:

3) Table 2: perhaps substitute ‘best fit’ with ‘% subjects with best fit’ to make it clearer to the reader?

Response to Reviewer #1, comment 9.3:

We agree with the reviewer that reporting the relative proportions of subjects that were fitted best by each model would enhance comprehension and added this relative proportion to Table 2.

Reviewer #1, comment 9.4:

4) Line 464: the authors could consider citing Buabang et al 2023 (doi: 10.1037/xge0001280) for empirical data supporting the point on inaccurate representation of action-outcome contingencies during extinction.

Response to Reviewer #1, comment 9.4:

We thank the reviewer for this suggestion and added the respective reference (line 861).

“However, a lack of goal-directed behavior can alternatively emerge due to an inaccurate representation of action-outcome contingencies during extinction (Buabang et al., 2023), or random responding due to a lack of motivation (Watson et al., 2022).”

Reviewer #1, comment 9.5:

5) I think there is a typo in the legends of Figure 11C: PMC instead of PMS.

Response to Reviewer #1, comment 9.5:

We thank the reviewer for reporting this typo. We corrected the legend of Figure 5C.

Reviewer #1, comment 9.6:

6) I didn't understand 'However, the repetition bias could also decrease the probability to switch back to the DAS when the DAS is optimal later during the block.' Perhaps some rephrasing is needed?

Response to Reviewer #1, comment 9.6:

Yes certainly, we rephrased the sentence to make the point clearer (line 504).

“However, if goals shift into the range where the DAS is one of the optimal sequences (e.g., trials 7 to 10 of goal trajectory 4, see Figure 8B), a strong repetition bias for specific sequences can decrease the probability that participants switch back to the again-optimal DAS.”

Reviewer #1, comment 9.7:

7) Similarly, I also did not understand the sentence on lines 211 – 213: 'we calculated the correlation between the difference of used sequences and the difference of mean reward per trial between the first and the second half.'

Response to Reviewer #1, comment 9.7:

We rephrased the respective sentence (line 557):

“To assess if this decrease in behavioral variability was caused by learning to find rewarding sequences more easily, we calculated the correlation of (1) the difference in the number of used sequences between the first and second half of the experiment with (2) the difference of the mean reward per trial between the first and the second half.”

Reviewer #2:

We thank the reviewer for his/her valuable and constructive advice when reviewing our manuscript. We addressed all the comments.

Note that line numbers refer to the new manuscript file **without** marked changes. We added two versions, one with, and the other without marked changes.

Reviewer #2, initial comment:

The authors report a study of repetition bias using a newly developed sequential decision-making task combined with computational modeling of choice behavior. They find evidence for a repetition bias in participants' behavior in addition to reward-oriented, goal-directed choice strategies, but also report inter-individual differences in how strongly participants rely on past choice frequency to inform their decisions. They discuss their findings in the context of habit learning.

The manuscript is timely and relevant, well written, clear, and structured. The research question and hypotheses are well informed by the literature and the methods are sound and well-chosen to answer the question at hand. The description of the computational models used are clear and comprehensible. I want to commend the authors for openly sharing their data, materials, and analysis code. I have read the paper with enthusiasm and offer only a few minor points I would like to invite the authors to address.

Reviewer #2, comment 1:

1. One problem of previous experimental approaches to the induction and measurement of frequency-based behavioral effects is that often the magnitude of reinforcement and choice frequency are correlated by design. Thus, participants have an incentive to choose options more often leading to more reward. You seem to have mitigated this effect by your task design giving participants instead an incentive to reach a specified, changing level of reward, yet I would like you to discuss this point more explicitly in your manuscript drawing more attention to this issue.

Response to Reviewer #2, comment 1:

We thank the reviewer for this suggestion. We added the following paragraph to the *Experimental Task* Section (manuscript file without marked changes, line 171):

“Note that most studies that investigate the influence of repetition on value-based decisions use options with either fixed rewards (e.g., Hardwick et al., 2019; Nebe et al., 2024), or fixed rewards combined with changing reward probabilities (e.g., Daw et al., 2011; Frölich et al., 2023). In our task, through changing goals and grid layouts, we transparently vary the (relative) reward of the different options. With this, repeating an option throughout the task is not optimal. This results in a lower correlation between reward magnitude and behavioral repetition.”

Reviewer #2, comment 2:

2. The Participants section of your Methods does not state how you decided on the aimed-at sample size. Did you perform a sample size analysis prior to data assessment? If so, please

state this explicitly in the Methods. If not, please provide your thoughts in the process of determining the sample size.

Response to Reviewer #2, comment 2:

Yes, we did not perform sample size calculations, and we added this information and the reason for not performing prior sample size calculations to the *Participants* section (line 104). We used a newly developed task in combination with a new computational model, and therefore could not estimate expected effect sizes from existing literature for the model-free analyses. To ensure that with our sample size we should be able to detect effects, our main analysis is based on Bayesian statistics, where we used parameter recovery and posterior predictive checks instead of a sensitivity analysis to validate our model.

“We did not perform sample size calculations for our model-free analyses prior to data collection, as we used a newly developed task in combination with a new computational model, and therefore could not estimate expected effect sizes from existing literature. We used parameter recovery and posterior predictive checks to validate our model-based analyses.”

Reviewer #2, comment 3:

3. The sample is imbalanced regarding gender (50 female out of 70 participants). I would not expect sex to have an influence on this basic learning process, yet a discussion of this point (and other sample size characteristics like the ethnicity, socioeconomic status, and education level of participants) would be valuable for the later integration and generalization of your findings.

Response to Reviewer #2, comment 3:

We thank the reviewer for bringing this to our attention. We calculated the descriptive statistics depending on gender and added the results to *Behavioral Results* section (line 457). Unfortunately, we did not collect data on other characteristics (ethnicity, SES, and education) besides gender and age.

“Furthermore, female and male participants did not differ significantly in their propensity to use the DAS (female: $M = .53, SD = .19$; male: $M = .57, SD = .17$; Welch- t -test, $t(39.54) = -0.81, p = .421, d = .20, CI = [-0.13, 0.06]$). Therefore, we could not find gender-related differences in the propensity to use the proposed DAS. However, male participants achieved a significant higher mean reward per trial than female participants (female: $M = 79.89, SD = 5.93$; male: $M = 82.69, SD = 5.79$; Mann-Whitney U test, because the assumption of normality was violated, $U = 302.00, p = .010, CL = 0.30$). Furthermore, male participants were significantly faster than female participants (female: $M = 1,755.47ms, SD = 415.56ms$; male: $M = 1,481.13ms, SD = 274.59ms$; Mann-Whitney U test, because the assumption of normality was violated, $U = 723.00, p = .004, CL = 0.72$), and male participants had significantly fewer time outs (female: $M = 5.32, SD = 3.89$; male: $M = 3.30, SD = 2.74$; Mann-Whitney U test, because the assumption of normality was violated, $U = 654.50, p = .044, CL = 0.65$). Of these results, only the differences in the proportion of DAS choices would have direct relevance to our results. As this

difference was non-significant, we do not further analyze the differences in the other performance measures.”

Reviewer #2, comment 4:

4. The correlational analyses you present in Figures 3C and 6 are probably underpowered (Kretzschmar & Gignac, 2019; Schönbrodt & Perugini, 2013) and, at least in the case of the results in Fig. 3C, seem influenced by a few influential data points in the upper left quadrant. Please discuss this shortcoming in terms of statistical power and use caution when interpreting the correlational effects.

Response to Reviewer #2, comment 4:

We agree with the reviewer and added the following discussion on the potential limitation of these correlations at *Decrease in behavioral variability* (line 562) and *Correlations between parameters of EVPRM* (line 707).

“This correlation showed a significant relationship in the expected direction, $r = -.38, p = .001, CI = [-.56, -.15]$ (see Figure 3C), but could have been driven by few participants with high increases in reward in combination with a high decrease in used sequences. Despite this potential limitation, our analysis suggests that a decrease in behavioral variability was mainly associated with an increase in mean rewards, as more participants with a decrease in behavioral variability improved their mean rewards. This in turn suggests that the decrease in behavioral variability could be caused by learning to identify sequences with high rewards, over the course of the experiment.”

“We analyzed the correlations between the parameter estimates for the three free model parameters repetition bias h , precision on expected rewards β , and approximated reward R_0 . As these correlations are potentially not stable with our sample size (Schönbrodt & Perugini, 2013), we use this analysis only as preliminary evidence for the expected tendencies to further validate our model.”

Reviewer #2, comment 5:

5. The 27 participants best described by the EVPRM seem to start with an especially low proportion of DAS choices (46.2% choice of the DAS in the first half to 51.6% in the second half (ll. 324-326) vs. 53.3% in the first half to 55.2% in the second half for the whole sample; ll. 146-148). In how far might the results of your model-free and model-based analyses be influenced by differences in starting point rather than an increased tendency to repeat DAS? Might this point towards difference in how participants approach the task or understand the instructions even before behavioral frequency can have an effect on behavior?

Response to Reviewer #2, comment 5:

We thank the reviewer for bringing up this point. Our model-free analyses based on summary measures were indeed affected by differences in initial $p(\text{DAS})$ across participants. For instance, when we tested the difference of $p(\text{DAS})$ between the first and second half of the experiment. Here initial $p(\text{DAS})$ certainly affected the aggregate statistics that were used for this analysis.

In contrast, model fitting for all models should be unaffected by the DAS proportion during the first half of the experiment, as the effects of repetition-based and goal-directed components are calculated on a trial-by-trial basis and do not incorporate aggregates on probabilities (for DAS and non-DAS choices). Furthermore, all models we considered can produce $p(\text{DAS})$ in the entire range observed in our participant data, depending on the model parameters. Therefore, the lower $p(\text{DAS})$ during the first half of the experiment of the EVPRM group should not influence model fitting or model comparison. An exception to this would be very high values of $p(\text{DAS})$, which would cause posterior distributions over parameters to be very wide. This is because in this case, a ceiling effect on any possible increases of $p(\text{DAS})$ makes DAS trials uninformative to model fit and comparison. In addition, non-DAS trials, which could be informative, would be very scarce. This is one of the reasons we excluded participants with $p(\text{DAS}) > .90$, as their choices were not informative for model fitting and comparison.

However, to investigate a potential influence of initial $p(\text{DAS})$ on the repetition bias parameter h , we split participants into low and high initial $p(\text{DAS})$ and tested for differences of the repetition bias parameter h . This new analysis did not yield a significant difference across groups. This points towards the fact that the distribution of repetition bias strength uncovered by our model goes beyond initial $p(\text{DAS})$ differences. We now report this new analysis in the section *Increase in DAS usage in participants fitted best with EVPRM* (line 701):

“Furthermore, we tested if participants below the median of initial proportion of DAS choices have a greater repetition bias strength h compared to participants with initial $p(\text{DAS})$ above the median. We could not find a significant difference using a two-sample t -test, $t(68) = -0.996, p = .322, d = 0.24, CI = [-0.27, 0.09]$. This means that we could not find evidence that the differences in repetition bias strength found by EVPRM are dependent of initial $p(\text{DAS})$.“

Given our model-based results, particularly the difference in initial $p(\text{DAS})$ mentioned by the reviewer, we speculate that participants with a high repetition bias (as we found with the EVPRM group) might be less inclined to engage in strongly goal-directed behavior. If this is the case, their behavior during the first few trials might be less goal-directed and more exploratory than for other participants, lowering the initial $p(\text{DAS})$ during the first few trials. As trials pass, given their high repetition bias, the fact that high expected rewards of the DAS were indirectly communicated, and they likely received lower rewards compared to the DAS, $p(\text{DAS})$ increases more steeply in the EVPRM group than in other participants. This increase, still within the first half of the experiment, might mask an initially low $p(\text{DAS})$, leading to a $p(\text{DAS})$ over the entire first half not that much lower than the non-EVPRM group (46.2% vs 53.3%). However, the formulation of the EVPRM model, where all trials contribute to the inference, does not allow to directly test the influence of different initial developments of $p(\text{DAS})$ on the repetition bias parameter. We have added this point to the *Limitations* section of the manuscript (line 877):

“Consistent with model assumptions, participants best fitted by EVPRM showed stronger repetition biases and an increase of DAS choice proportions from the first to the second half of the experiment. Note that this group also had a below-average

proportion of DAS choices during the first half of the experiment (46.2% vs 53.3%). One potential explanation for this finding is that these participants with high repetition bias strength were in general less goal-directed and made more exploratory choices during the initial trials of the task. Subsequently, based on their strong repetition bias and after learning that there is a high proportion of trials where the DAS was one of the sequences with the highest expected reward, they increased their DAS choice proportions more steeply than other participants. However, the design of our task does not allow us to directly test this hypothesis.”